# VISUAL SELF-REFINE: A PIXEL-GUIDED PARADIGM FOR ACCURATE CHART PARSING

**Jinsong Li**[1,2*] **Xiaoyi Dong**[1,2†] **Yuhang Zang**[2] **Yuhang Cao**[2] **Jiaqi Wang**[2,4†] **Dahua Lin**[1,3]
[1] The Chinese University of Hong Kong   [2] Shanghai AI Laboratory
[3] CPII under InnoHK   [4] Shanghai Innovation Institute
lj024@ie.cuhk.edu.hk
Github: https://github.com/InternLM/VSR

## ABSTRACT

While Large Vision-Language Models (LVLMs) have demonstrated remarkable capabilities for reasoning and self-correction at the textual level, these strengths provide minimal benefits for complex tasks centered on visual perception, such as Chart Parsing. Existing models often struggle with visually dense charts, leading to errors like data omission, misalignment, and hallucination. Inspired by the human strategy of using a finger as a "visual anchor" to ensure accuracy when reading complex charts, we propose a new paradigm named Visual Self-Refine (VSR). The core idea of VSR is to enable a model to generate pixel-level localization outputs, visualize them, and then feed these visualizations back to itself, allowing it to intuitively inspect and correct its own potential visual perception errors. We instantiate the VSR paradigm in the domain of Chart Parsing by proposing ChartVSR. This model decomposes the parsing process into two stages: a Refine Stage, where it iteratively uses visual feedback to ensure the accuracy of all data points' Pixel-level Localizations, and a Decode Stage, where it uses these verified localizations as precise visual anchors to parse the final structured data. To address the limitations of existing benchmarks, we also construct ChartP-Bench, a new and highly challenging benchmark for chart parsing. Our work also highlights VSR as a general-purpose visual feedback mechanism, offering a promising new direction for enhancing accuracy on a wide range of vision-centric tasks.

## 1 INTRODUCTION

Large Vision-Language Models (LVLMs) have undergone rapid development (Bai et al., 2023b; Wang et al., 2024a; Bai et al., 2025), evolving from handling simple tasks to tackling complex and challenging ones. The emergence of models with "thinking" capabilities (Wei et al., 2022), such as OpenAI o1 (Jaech et al., 2024) and Gemini-2.5-Pro (Comanici et al., 2025), allows them to identify and rectify errors present in their direct responses, thereby achieving superior performance. However, their reasoning and deliberation processes are predominantly conducted at the textual level. This paradigm is highly effective for purely text-based tasks, such as mathematical problem-solving (Muennighoff et al., 2025; Chen et al., 2025), where textual feedback is well-suited for reflection and error correction. In contrast, for tasks where the core challenge lies in "visual perception", such as Chart Parsing, this text-centric self-correction yields minimal benefits, as illustrated in Figure 2. This observation motivates us to explore a critical question: for tasks centered on visual perception, can we introduce "visual feedback" to enable an effective form of visual reflection and refinement?

The Chart Parsing task serves as a prime example of a "visual perception" heavy challenge. As a crucial visual medium for presenting structured information, charts contain data that is often precise and critical, demanding accurate extraction and utilization. Even minor errors in data extraction can have a significant impact on downstream tasks. Early Chart Parsing methods relied heavily on OCR and hand-crafted rules (Savva et al., 2011; Jung et al., 2017), which struggled to generalize to diverse chart types. More recently, approaches in the era of LVLMs typically train end-to-end on data pairs of chart-annotation format. While these models achieve promising results on simple

---

*This work is done during an internship in Shanghai AI Laboratory. † Corresponding author

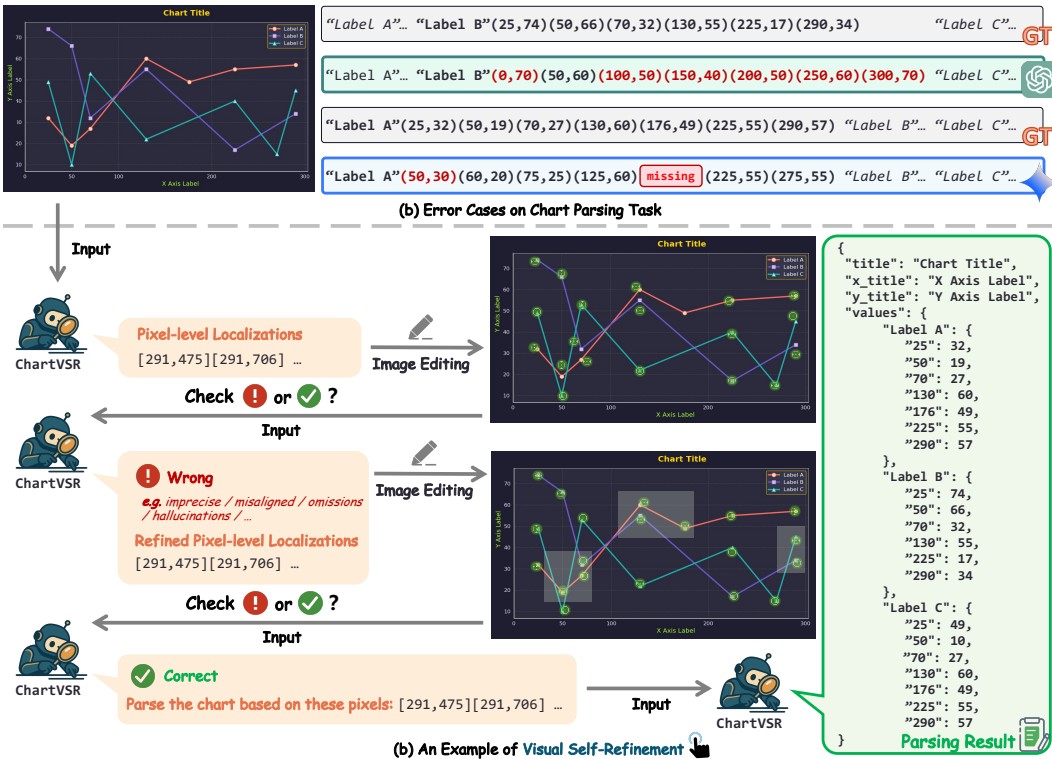

Figure 1: **Overview of Chart Parsing and Visual Self-Refine (VSR).** (a) Even strong models often fail to produce entirely correct results in a single pass of Chart Parsing. (b) A schematic illustration of the proposed VSR method. The process follows the sequence indicated by the *gray arrows*.

charts, their performance degrades significantly on charts with high visual information density and no explicit numerical labels (Han et al., 2023; Xia et al., 2024; Chen et al., 2024). They frequently exhibit issues such as large numerical errors, misaligned data correspondences, data omissions, and even hallucinations of non-existent data points. (Zhang et al., 2025; Liu et al., 2025)

For charts with high visual density and no explicit numerical labels, such as the one shown in Figure 1 (a), even strong closed-source models like GPT and Gemini perform unsatisfactorily, with their responses containing multiple errors. For humans, however, while meticulously extracting all information from such a chart might be a slow process requiring careful inspection, it is a straightforward task. Although humans find it challenging to read such charts accurately at a single glance and are prone to misreading or confusion, a common human strategy is to use a finger to point at each data point sequentially. The position of the finger acts as a "visual anchor", enabling more accurate value readings while effectively preventing issues like reading duplicate points, omitting points, or misaligning correspondences.

Inspired by this human strategy, we propose a new paradigm named Visual Self-Refine (VSR). The core idea is for a model to generate pixel-level localization outputs, visualize them, and then feed these visualizations back to itself as visual input. This process allows the model to see its own previous work, giving it an opportunity to identify and correct potential visual perception errors such as mislocalizations, omissions, or hallucinations.

We instantiate VSR paradigm in the domain of Chart Parsing by proposing the ChartVSR. As shown in Figure 1 (b), for an input chart, the model first generates pixel-level localization outputs. These outputs are then visualized and fed back into the model as visual feedback for it to inspect. Compared to purely textual feedback, visual feedback allows the model to more intuitively identify and correct its prior visual perception results. Once the model confirms that the visualized output is correct, it uses these precise, "finger-pointing" locations to parse the final structured data from the chart.

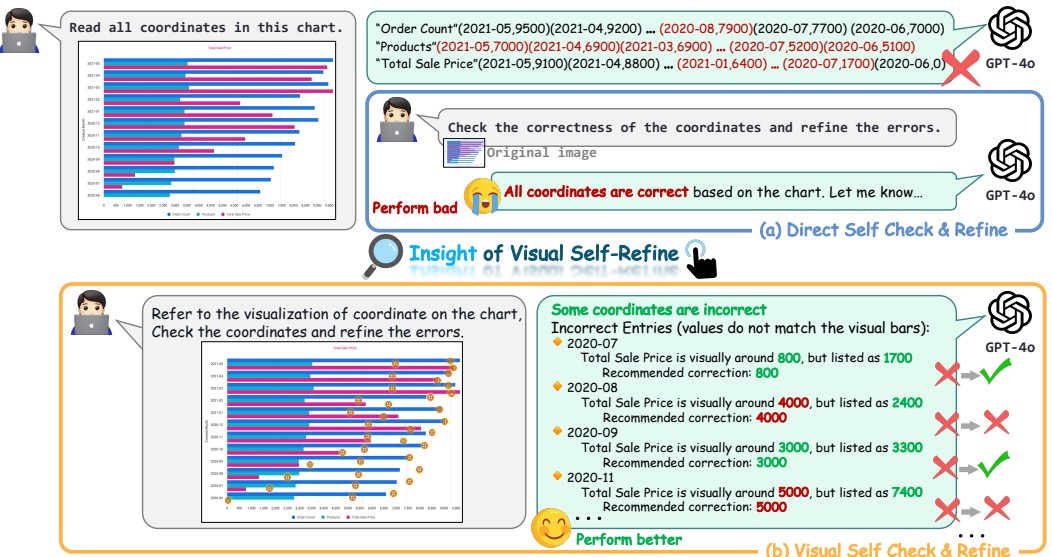

Figure 2: **Insight of Visual Self-Refine** (a) Without visual feedback, strong LVLM (GPT-4o) fails to identify its own parsing errors. (b) In contrast, when the model's output is visualized onto the chart (yellow markers). With this explicit visual feedback, the same model can now readily spot discrepancies and identify its mistakes. This highlights the ineffectiveness of direct self-correction for vision-centric tasks and motivates our proposed Visual Self-Refine approach.

Furthermore, beyond existing Chart Parsing benchmarks, we have constructed ChartP-Bench, a new and highly challenging benchmark, created by manually selecting and annotating complex charts from multiple data sources. Crucially, VSR, as a visual feedback paradigm, is not limited to Chart Parsing. We also demonstrate its potential applicability on other tasks, including Visual Counting and Visual Grounding. In summary, our contributions are threefold:

a) We introduce the concept of Visual Self-Refine (VSR), a novel paradigm that utilizes visual feedback for self-correction, offering a promising new direction for tackling vision-centric tasks.

b) We instantiate the VSR paradigm in the Chart Parsing domain with our ChartVSR model, experimentally demonstrating the paradigm's effectiveness in improving parsing accuracy.

c) We present ChartP-Bench, a high-quality and challenging benchmark, to foster future research in the Chart Parsing domain.

## 2 METHODS

### 2.1 CHART PARSING

Chart Parsing is a fundamental visual perception task aimed at extracting structured numerical and textual information from a chart image, such as converting a line plot into its underlying data json. This process is a crucial step for enabling precise downstream reasoning, like chart-based question answering. Early work such as DePlot (Liu et al., 2022a) framed the task as plot-to-table generation. ChartVLM (Xia et al., 2024) enhance performance through supervised fine-tuning on chart-specific datasets. OneChart (Chen et al., 2024) introduce numerical loss to explicitly supervise numerical accuracy. Existing models still struggle with high-visual-density charts, frequently exhibiting data omission, hallucination, or numerical deviations. Further related works are provided in Appendix C.

### 2.2 CHARTVSR

Our ChartVSR model is built upon Qwen2.5-VL-3B (Bai et al., 2025), an open-source LVLM. Its architecture consists of three components: a Vision Transformer based vision encoder, a Qwen2.5 language model as the reasoning core, and an MLP-based merger to connect them. The vision encoder divides the input image into $28 \times 28$ non-overlapping visual patches and generates dense

visual embeddings that preserve the spatial granularity essential for Chart Parsing. We finetune this model using a specialized data curation and training pipeline (detailed in Section 3).

## 2.3 VISUAL SELF-REFINE FOR CHART PARSING

The core of ChartVSR is the Visual Self-Refine (VSR) paradigm, which decomposes the traditional one-shot parsing process into two logical stages: the **Refine Stage** and the **Decode Stage**. This design mimics the human behavior of using a finger to aid in reading a chart, aiming to ensure perceptual accuracy through explicit visual feedback.

**Refine Stage: Iterative Refinement via Visual Feedback**  The goal of this stage is for the model to "point out" all key data elements in the chart, much like a human using a finger. We provide the model with the original chart image. The model's task is not to read the values directly, but to output a list of **Pixel-level Localizations** (a series of [x, y] pixel coordinates corresponding to each data point).

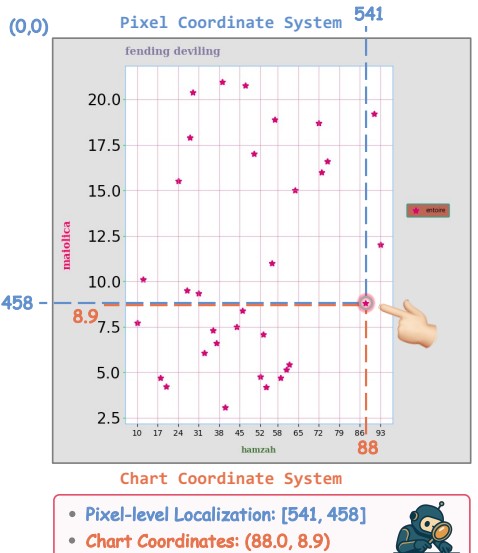

Figure 3: **An illustrative case of ChartVSR.**

This key step deconstructs the complex chart parsing task into a simpler visual grounding problem, where the model can focus solely on "where" the data points are, without the immediate need to parse "what" they are or "how much" they represent. The generated Pixel-level Localizations serve as intermediate, structured visual anchors. Subsequently, we visualize these Pixel-level Localizations on the original chart image using predefined markers. This edited image is then fed back into the model as explicit **visual feedback**. By observing its own previously generated markers, the model can identify imprecise localization, omissions, or hallucinations, and subsequently output a set of corrected Pixel-level Localizations. This "generate-feedback-correct" loop can be performed iteratively until the model indicates that the visualize localizations are correct or a preset maximum number of iterations is reached.

**Decode Stage: Parsing from Visual Anchors**  Once the Refine Stage is complete, we obtain a set of high-precision Pixel-level Localizations that have been verified by the model itself. In the Decode Stage, the model receives the original chart image along with this final set of Pixel-level Localizations as input. At this point, the model's task shifts from "localization" to "interpretation".

Guided by these precise visual anchors, the model correlates them with the chart's visual elements. For instance, as shown in Figure 3, for a data point on a bar chart, the model utilizes its Pixel-level Localization [541, 458] to infer its value within the chart's coordinate system (i.e., **Chart Coordinates** (88.0, 8.9)) and associates it with the corresponding legend label. By decoupling the two sub-tasks of perception (localization) and parsing (interpretation), and using Pixel-level Localizations as the bridge, ChartVSR effectively mitigating common errors such as data misalignment, omissions, and numerical deviations.

## 3 TRAINING DATA

### 3.1 ANALYSIS OF EXISTING CHART DATASETS

Existing chart datasets can generally be divided into two categories: **(1) non-parsable QA datasets** and **(2) fully annotated parsable datasets**. Non-parsable Question Answering (QA) datasets (Xu et al., 2023; Wang et al., 2024b) typically present tasks in a question-based format that probes specific aspects of a chart. Consequently, they require only partial chart-related information to answer the questions, rather than a full parsing of the chart's entire contents. This characteristic allows

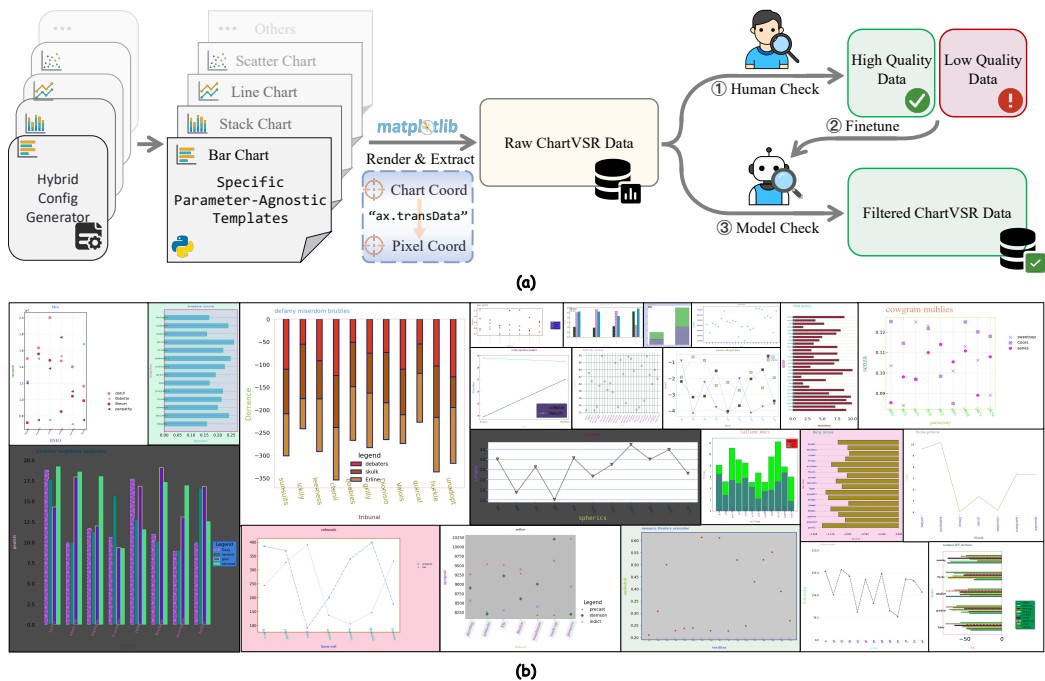

Figure 4: **Overview of our Data Engine.** (a) The ChartVSR data generation pipeline. (b) Example data generated by our data engine, showcasing high diversity and high quality.

them to be conveniently collected from publicly accessible sources such as websites and academic papers. In contrast, parsable chart datasets (Masry et al., 2022; Methani et al., 2020; Xia et al., 2024) necessitate comprehensive annotations, such as the original drawing code, tables, JSON, or CSV files corresponding to the charts. Such detailed annotations are rare in real-world sources, making it difficult to acquire large-scale real-world data. Consequently, almost all parsable chart datasets are synthesized using plotting tools such as Matplotlib, Pyecharts, or MATLAB. Existing synthesized parsable datasets commonly suffer from limited diversity, implicit regularities, and data homogeneity.

a) **Lack of Stylistic Variety:** Existing datasets typically apply adjustments to only a few visualization parameters (e.g., color, pattern, or fonts), resulting in similar visual styles across generated charts. This stylistic homogeneity significantly hinders the model's generalization capability, leading to poor performance when applied to visually diverse real-world charts.

b) **Implicit Regularities and Data Homogeneity:** Many existing parsable datasets are sourced from the web or specific statistical databases. These datasets often contain monotonic trends, recurring strings, or implicit regularities (e.g., specific company names frequently co-occur, or numeric labels around the year 2000 typically represent incremental year sequences). As shown in Appendix D, models trained on them often overfit to these patterns, showing fast convergence but poor real-world generalization, leading to hallucinations and degraded performance.

## 3.2 DATA ENGINE

To overcome these limitations, we meticulously designed and developed a highly diverse, robust, and scalable data engine. This engine consists of two core components: *Parameter-Agnostic Templates* and *Hybrid Configuration Generators*. As illustrated in Figure 4 (a), for each chart type, we first construct a Parameter-Agnostic Template, which is a universal plotting script capable of generating both a chart and its corresponding annotations simultaneously. These templates comprehensively cover all relevant plotting parameters available in Matplotlib for the specific chart types.

Building upon these templates, we developed a Hybrid Configuration Generator that automatically produces a wide range of diverse chart configurations. It samples across Matplotlib's full visualization space, covering dimensions such as color schemes, fonts, and layouts. To synthesize chart elements

(e.g., labels, titles, data points), it combines real-world content with randomly generated content. This process maximizes stylistic diversity and avoids the implicit regularities common in existing datasets, thereby promoting a more robust Chart Parsing capability in the model. To quantitatively demonstrate the superiority of our data, we propose four key metrics for evaluating dataset quality. As shown in Table 1, our dataset excels on these metrics. (Detailed calculation methods and explanations for each metric are in Appendix E)

In the initially generated dataset, we observed that a small portion of charts suffered from poor visual quality due to unusual layout configurations or suboptimal combinations of plotting parameters produced by the Hybrid Configuration Generator. To address this, we manually reviewed and categorized charts of various types, identifying and labeling low-quality samples. Based on these positive and negative examples, we finetuned a Qwen2-VL-2B model

Table 1: Quantitative comparison across four chart datasets. (↑) indicates higher is better, while (↓) indicates lower is better. Best values are in **bold** and second-best are underlined.

| Metric | Ours | ChartQA | PlotQA | ChartX |
|---|---|---|---|---|
| Avg. Data Points/Chart (↑) | **22.69** | 15.04 | 12.48 | 10.95 |
| Unique Numerical Ratio (↑) | **59.20%** | 22.77% | 50.79% | 4.06% |
| Avg. Abs. Correlation (↓) | **0.38** | 0.76 | 0.77 | 0.71 |
| Avg. PMI of Top-K Pairs (↓) | **6.14** | 6.39 | 6.66 | 8.61 |
| User Study (4-point) (↑) | **3.82** | 3.15 | 2.68 | 2.61 |

(Wang et al., 2024a) to serve as a data filter. This model was then applied to the raw dataset to discard low-quality charts. Through this entire pipeline, we constructed a final high-quality dataset containing approximately 800K samples.

## 3.3 TRAINING DATA FORMAT

The format is designed to align with the two stages of ChartVSR. For the **_Refine Stage_**, we construct three types of training samples to teach the model to: a) generate locations from scratch, b) correct erroneous markers (e.g., missing, shifted, or hallucinated), and c) confirm correct markers to terminate the refinement loop. For the **_Decode Stage_**, the objective is to train the model to parse structured data, given accurate visual anchors. A training sample consists of the original chart image and a set of verified-as-accurate pixel-level coordinates. The model's task is to parse the chart into a structured JSON object containing numerical values and metadata, guided by these precise visual anchors.

## 4 EXPERIMENTS

### 4.1 IMPLEMENTATION DETAILS

We conduct our training on a server equipped with 8 NVIDIA H200 GPUs, each with 144GB memory. To prevent mismatches between our Pixel-level Localizations and the model's internal coordinate system due to resizing, we resize the longer side of each input image to 1036 pixels, a multiple of 28. The model is trained for one epoch with a total batch size of 128. We use the AdamW optimizer with a learning rate of $2 \times 10^{-7}$, a warmup ratio of 0.03, and a weight decay of 0.01. All three components of the model are unfrozen and trained end-to-end.

### 4.2 BENCHMARKS AND METRICS

**Existing Benchmarks**   Although many existing benchmarks are related to charts, most focus on Chart Question Answering tasks. Only a few include fully annotated, parsable chart structures suitable for evaluating Chart Parsing, such as ChartQA-SE (Chen et al., 2024; Masry et al., 2022), PlotQA-SE (Chen et al., 2024; Methani et al., 2020), and ChartX-SE (Chen et al., 2024; Xia et al., 2024). ChartQA-SE contains real-world charts with natural layouts and visual designs; PlotQA-SE consists of charts synthesized from real-world data; and ChartX-SE is fully synthetic, rendered using Matplotlib. All three benchmarks are converted from Chart Question Answering datasets, and some contain explicit numeric annotations on the chart images.

During a manual spot-check on ChartQA-SE, we discovered several mislabeled samples, which prompted a full inspection of all 1,509 test samples. We found that over 100 samples had incorrect ground-truth annotations (see Appendix G for details). To ensure evaluation reliability, we removed these erroneous cases and created a cleaned subset, named ChartQA-SE-Clean.

Table 2: **Results on general Chart Parsing benchmarks.** We report Standard SCRM Average Precision (AP): **Strict** ($J_{thr} = 0$, $e_{thr} = 0$), **Slight** ($J_{thr} = 2$, $e_{thr} = 0.05$), and **High** ($J_{thr} = 5$, $e_{thr} = 0.10$). The best results are highlighted in **bold** and second-best are underlined. Model sizes are listed for fair comparison.

| Model | Size | ChartQA-SE-Clean | | | PlotQA-SE | | | ChartX-SE | | |
|---|---|---|---|---|---|---|---|---|---|---|
| | | AP-Strict | AP-Slight | AP-High | AP-Strict | AP-Slight | AP-High | AP-Strict | AP-Slight | AP-High |
| DePlot | 1.3B | 55.61 | 66.89 | 70.89 | 3.11 | 16.49 | 26.50 | 14.84 | 33.42 | 42.42 |
| ChartAst | 13B | 33.81 | 63.82 | 71.23 | 5.18 | 48.67 | 56.08 | 18.78 | 36.76 | 47.28 |
| ChartVLM | 7.3B | 65.94 | 77.17 | 82.11 | 3.81 | 46.83 | 54.00 | 25.42 | 34.92 | 40.82 |
| OneChart | 0.2B | **66.20** | 78.89 | 83.92 | 34.56 | 84.18 | 86.10 | **44.55** | 53.12 | 59.72 |
| ChartVSR | 3B | 65.70 | **83.69** | **85.64** | **34.99** | **84.61** | **88.10** | 43.99 | **53.48** | **62.89** |

Table 3: **Results on ChartP-Bench.** The benchmark is divided into an **Easy subset** (charts with $\leq 18$ data points) and a **Hard subset** (charts with $>18$ data points). We report Standard SCRM Average Precision (AP): **Strict** ($J_{thr} = 0$, $e_{thr} = 0$), **Slight** ($J_{thr} = 2$, $e_{thr} = 0.05$), and **High** ($J_{thr} = 5$, $e_{thr} = 0.10$). The best results are highlighted in **bold** and second-best are underlined. Model sizes are included for fair comparison.

| Model | Size | Easy Subset | | | | Hard Subset | | | | Avg. |
|---|---|---|---|---|---|---|---|---|---|---|
| | | AP-Strict | AP-Slight | AP-High | Avg. | AP-Strict | AP-Slight | AP-High | Avg. | |
| *Strong Closed-source LVLMs* | | | | | | | | | | |
| GPT-4o | - | 0.00 | 1.69 | 6.73 | 2.81 | 0.00 | 0.99 | 4.14 | 1.71 | 2.09 |
| Gemini-2.5-Flash | - | 0.00 | 37.00 | 55.57 | 30.86 | 0.00 | 23.84 | 40.22 | 21.35 | 24.62 |
| Gemini-2.5-Pro | - | 0.06 | 36.51 | 46.05 | 27.54 | 0.00 | 46.11 | 66.39 | 37.50 | 34.07 |
| *Chart LVLMs* | | | | | | | | | | |
| DePlot | 1.3B | 0.00 | 9.73 | 15.35 | 8.26 | 0.00 | 1.27 | 4.19 | 1.82 | 4.04 |
| ChartAst | 13B | 0.00 | 1.56 | 5.79 | 2.45 | 0.00 | 0.13 | 3.10 | 1.08 | 1.55 |
| ChartVLM | 7.3B | 0.00 | 1.74 | 6.34 | 2.70 | 0.00 | 0.48 | 3.56 | 1.35 | 1.81 |
| OneChart | 0.2B | 0.00 | 4.75 | 18.89 | 7.88 | 0.00 | 1.98 | 9.48 | 3.82 | 5.22 |
| ChartVSR | 3B | **0.11** | **51.95** | **67.44** | **39.83** | **0.02** | **49.30** | **63.67** | **37.66** | **38.41** |

**ChartP-Bench** (*Chart Parsing Benchmark*)    To address the limitations of existing benchmarks, we introduce ChartP-Bench, a new and highly challenging benchmark for chart parsing. We manually selected, filtered, annotated, and cleaned 1,200 high-quality chart images from multiple sources. This benchmark is characterized by its high difficulty and visual information density, with each chart containing over 20 data points on average. Unlike existing datasets, ChartP-Bench has undergone rigorous quality control to ensure there are no annotation errors and to avoid issues such as stylistic homogeneity and implicit regularities.

**Evaluation Metrics**    We adopt the Structuring Chart-oriented Representation Metric (SCRM) (Xia et al., 2023) for evaluation. SCRM quantifies the structural consistency between a model's parsed result and the ground truth at both the image and dataset levels. It first computes the Intersection over Union (IoU) between predicted and ground-truth data triplets based on a predefined string similarity threshold, $J_{thr}$, and a relative numerical error threshold, $e_{thr}$. The overall performance is then evaluated by calculating the Average Precision (AP) over a range of IoU thresholds. For more details on SCRM, please refer to Appendix F.

### 4.3    MAIN RESULTS

**ChartVSR demonstrates robust and compelling performance on existing Chart Parsing benchmarks.**    As shown in Table 2, ChartVSR achieves competitive performance across three existing Chart Parsing benchmarks. It is noteworthy that these benchmarks vary in data origin and style: ChartQA-SE-Clean is sourced from real-world charts, PlotQA-SE is synthesized from real-world data, and ChartX-SE is entirely synthetic. The excellent results of ChartVSR across these datasets with different distributions, particularly under the more lenient Slight and High settings, demonstrate the strong generalization capabilities of our proposed method.

Table 4: **Ablation study on ChartVSR components.** The benchmark is divided into an **Easy subset** (charts with ≤18 data points) and a **Hard subset** (charts with >18 data points). We report Average Precision (AP) under three evaluation criteria: **Strict** ($J_{thr} = 0$, $e_{thr} = 0$), **Slight** ($J_{thr} = 2$, $e_{thr} = 0.05$), and **High** ($J_{thr} = 5$, $e_{thr} = 0.10$). The best results are highlighted in **bold**.

| Model | Easy Subset | | | | Hard Subset | | | | Avg. |
|---|---|---|---|---|---|---|---|---|---|
| | AP-Strict | AP-Slight | AP-High | Avg. | AP-Strict | AP-Slight | AP-High | Avg. | |
| ChartVSR | **0.11** | **51.95** | **67.44** | **39.83** | **0.02** | **49.30** | **63.67** | **37.66** | **38.41** |
| w/o VSR | 0.10 | 50.83 | 66.67 | 39.20 | 0.01 | 47.50 | 57.93 | 35.14 | 36.54 |
| w/o VSR & w/o Pixel | 0.10 | 50.61 | 66.19 | 38.97 | 0.00 | 46.63 | 57.50 | 34.71 | 36.17 |

Table 5: **Per-round analysis of VSR's error detection and correction capability.** We report the total number of charts with visual perception errors initially (Round 0) and after each refinement round. We also calculate the model's recall on errors from the previous round and its precision in confirming correct localizations.

Table 6: **VSR's computational cost.** The cost is measured by the total number of model forward inference calls, where $N_{max}$ denotes the maximum number of allowed refinement rounds.

| Refine Round | # Error Samples | Error Recall | Correct Confirmation |
|---|---|---|---|
| 0 | 110 | - | - |
| 1 | 51 | 92.3% | 88.2% |
| 2 | 54 | 85.5% | 76.0% |
| 3 | 52 | 85.8% | 76.6% |

| Method | # Inference Calls |
|---|---|
| Baseline (w/o VSR) | 1 |
| VSR ($N_{max} = 1$) | 3 |
| VSR ($N_{max} = 2$) | 3 to 4 |
| VSR ($N_{max} = n, n \geq 1$) | 3 to (n+2) |

**ChartVSR excels on the challenging ChartP-Bench.** On the more challenging ChartP-Bench, ChartVSR exhibits a significant performance advantage over other models, as detailed in Table 3. Among the powerful closed-source models, the Gemini series (especially Gemini-2.5-Pro) demonstrates strong capabilities, far surpassing other baselines, while GPT-4o struggles to handle this complex task (GPT-4o's outputs are provided in Supplementary Material). On both the Easy and Hard subsets, ChartVSR substantially outperforms all competing models, including the strongest Gemini model. As ChartVSR is a specialized model designed specifically for chart parsing, its superior performance over a general-purpose model like Gemini is expected. Furthermore, these results corroborate our earlier assertion that existing chart parsing models perform poorly on charts with high visual information density and stylistic diversity, which truly test visual perception capabilities. Furthermore, nearly all models score close to zero on the AP-Strict metric. This is because the numerical labels in ChartP-Bench are highly precise (e.g., 17.12 instead of integers like 10 or 20), making the strict requirement of zero numerical error ($e_{thr} = 0$) extremely demanding.

## 4.4 ANALYSIS ON VSR

**VSR significantly boosts chart parsing performance, especially on complex charts.** Table 4 shows that the full ChartVSR model achieves a clear performance gain over the version without the VSR module (w/o VSR). This improvement is particularly pronounced on the Hard subset, which contains more data points and is visually denser. This suggests that for simple charts, errors primarily stem from minor numerical deviations. For complex charts, however, the main perfor-

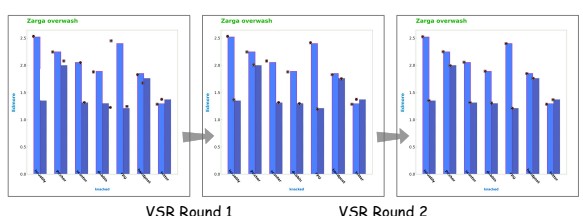

Figure 5: **A multi-round refinement case of ChartVSR.** The initial prediction (left) contains multiple errors, which are fully corrected after two rounds of refinement.

mance bottlenecks are structural errors like data point omissions or misalignments, and VSR's visual feedback mechanism is specifically designed to correct these visually salient errors. Furthermore, comparing the "w/o VSR" and "w/o VSR & w/o Pixel" variants reveals that merely introducing Pixel-level Localization without the accompanying VSR yields minimal performance gains.

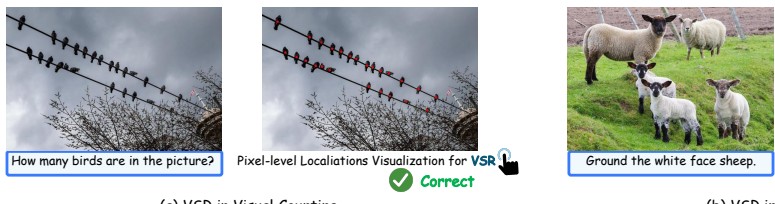

Figure 6: **The potential of the VSR paradigm on Visual Counting and Grounding.** (a) For Visual Counting, the model generates pixel-level localizations and uses the visualization to check for errors like omissions or duplicates. (b) For Grounding, the model visualizes its predicted bounding box to self-assess whether the localization is accurate and requires refinement. This illustrates the broad applicability of VSR as a general-purpose visual feedback mechanism for self-correction.

**VSR shows highly effective error correction in the first round, with diminishing returns subsequently.** As detailed in Table 5, the model successfully identifies 92.3% of initial errors in the first refinement round, reducing the number of error samples from 110 to 51 and correcting over half of the errors. In subsequent rounds, however, the total number of error samples does not decrease significantly, even though the model maintains an error recall rate of over 85%. This suggests that the remaining, more stubborn errors may stem from the model's deeper perceptual limitations, which are difficult to resolve through simple multi-round iteration. A potential direction for improvement could be to adopt a bootstrapping-like strategy: use the trained model for large-scale inference, collect cases where refinement fails, and add these hard cases to the training data for iterative finetuning. This could potentially enhance the model's ability to resolve complex errors in later rounds.

**VSR enhances performance at the cost of increased inference overhead.** As shown in Table 6, the VSR paradigm introduces additional forward inference calls. While a baseline model requires only a single call, VSR requires at least three (one for initial localization, one for visual feedback confirmation, and one for final decoding). This is a deliberate trade-off, analogous to the strategy employed by recent "thinking" large language models like OpenAI o1 and Gemini 2.5 Pro, which invest more computational resources during inference for deliberation and refinement to achieve higher accuracy and reliability.

### 4.5 POTENTIAL OF VSR ON OTHER TASKS

VSR is a general visual feedback paradigm, with applications extending beyond Chart Parsing. As illustrated in Figure 6, VSR can be seamlessly adapted to other tasks that demand high-fidelity visual perception, such as visual counting and visual grounding. For *visual counting*, a model can first output Pixel-level Localizations for all target objects. These localizations are then visualized on the original image and fed back to the model, giving it an opportunity to inspect for errors such as omissions or superfluous counts and then refine its answer. The application of VSR to *visual grounding* is even more natural, as the task's native output is a bounding box in pixel coordinates. The predicted bounding box can be drawn directly onto the image and returned to the model, allowing it to self-assess whether its localization is accurate and if the box requires adjustments in position or scale. VSR holds significant potential as a versatile tool for self-correction in vision-centric tasks.

## 5 CONCLUSION

In this work, we addressed the limitations of existing Large Vision-Language Models on visually intensive perception tasks like Chart Parsing by introducing Visual Self-Refine (VSR), a novel paradigm. Inspired by the human strategy of using a finger to aid in chart reading, VSR enables a model to generate pixel-level outputs, visualize them, and use this visualization as direct visual feedback to iteratively inspect and correct its own perceptual errors. We instantiated this paradigm with our ChartVSR model, which decouples the task into a Refine Stage and a Decode Stage, significantly improving the accuracy of data extraction. Our contributions are threefold: a) we proposed the general VSR paradigm as a new self-correction mechanism for vision-centric tasks; b) we implemented the ChartVSR model and experimentally demonstrated its state-of-the-art performance, especially on our new, challenging benchmark, ChartP-Bench, where it substantially outperforms

existing methods; and c) we introduced the high-quality ChartP-Bench to facilitate future research in this domain. Looking forward, the VSR paradigm holds significant potential for a broad range of other tasks requiring precise visual perception, such as visual counting and visual grounding. Future work could also explore methods to enhance the efficiency of the refinement process or leverage refinement failures as hard negatives for iterative model improvement.

ACKNOWLEDGMENTS

This project is funded in part by Shanghai Artificial Intelligence Laboratory, Shanghai Innovation Institute, the Centre for Perceptual and Interactive Intelligence (CPII) Ltd under the Innovation and Technology Commission (ITC)'s InnoHK. Dahua Lin is a PI of CPII under the InnoHK.

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

## A    REPRODUCIBILITY STATEMENT

To ensure the reproducibility of our work, we provide comprehensive details and resources in the main paper and the Supplementary Material. In Section 3, we have detailed the format of our training data, model implementation specifics, hyperparameter settings, and the full training pipeline. Furthermore, in the Supplementary Material, we have included key resources required to reproduce our core findings. These include: (1) ChartP-Bench, containing images and their corresponding ground-truth annotations; (2) visualization results from the ChartVSR inference process; (3) the example evaluation code for result assessment. We will open-source the data engine used for generating training data and model weights, upon the acceptance of this paper to facilitate further research within the community.

## B    THE USE OF LARGE LANGUAGE MODELS

We utilized Large Language Models (LLMs), such as Google's Gemini, as a writing assistance tool. The use of the LLM was strictly limited to improving the clarity, conciseness, and grammatical correctness of the text. Specific applications included proofreading for typographical errors, rephrasing complex sentences to enhance readability, and ensuring consistent terminology throughout the paper.

## C    RELATED WORK

**Chart-Specific Large Vision-Language Models.**    With the rapid advancement of large language models (LLMs) (Chiang et al., 2023; Touvron et al., 2023a; Yang et al., 2023; Team, 2023; OpenAI, 2023b; Zhang et al., 2024a; Ouyang et al., 2022; Chowdhery et al., 2022; Chen et al., 2024a;b), researchers have developed powerful multimodal systems by integrating visual encoders with language backbones and aligning them using large-scale image-text data (Chen et al., 2023). From early works such as CLIP (Radford et al., 2021), BLIP (Li et al., 2022), Instruct-BLIP (Dai et al., 2023) to more recent models like MiniGPT-4 (Zhu et al., 2023), LLaVA series (Liu et al., 2023a; 2024a;b), and Qwen-VL series (Bai et al., 2023b; Wang et al., 2024a; Bai et al., 2025), general-purpose large vision-language models (LVLMs) have achieved impressive performance across a wide range of vision-language tasks (Liu et al., 2023b; Fu et al., 2023; Yue et al., 2023; Xing et al., 2024). Concurrently, recent advancements have significantly enhanced LVLMs in complex visual perception (Zhang et al., 2023; Qian et al., 2024b; Zhang et al., 2024b; Qian et al., 2025; 2024a; Ding et al., 2024), establishing robust baselines for handling intricate visual details. However, applying these general LVLMs directly to chart-related tasks often leads to suboptimal results. To bridge this gap, several chart-specific LVLMs have been proposed, including ChartLLaMA (Han et al., 2023), ChartVLM (Xia et al., 2024), ChartPaLI (Carbune et al., 2024), and ChartAst (Meng et al., 2024), which rely on supervised fine-tuning using chart-specific datasets to align visual and linguistic modalities. More recently, ChartMoE (Xu et al., 2024) introduced a powerful mixture-of-experts model trained on aligned chart-text pairs in formats like JSON, tables, and CSV, achieving strong performance on multiple chart QA benchmarks. Overall, most existing chart-specific LVLMs focus primarily on high-level understanding tasks such as chart question answering (Masry et al., 2022; Methani et al., 2020; Xu et al., 2023; Wang et al., 2024b), while paying limited attention to the low-level fundamental task – chart perception.

**Chart Parsing Task.**    Chart Parsing involves extracting structured numerical and textual information from visual charts, a crucial step for enabling precise downstream reasoning. Early work DePlot (Liu et al., 2022a) framed the task as plot-to-table generation: a MATCHA (Liu et al., 2022b) is fine-tuned to parse the chart's underlying table. Similarly, ChartAST (Meng et al., 2024) also trains models directly on large-scale chart-to-table datasets. Recently, OneChart (Chen et al., 2024) proposes an integrated solution that enhances LVLMs with the number loss. By explicitly supervising numerical accuracy and using an auxiliary token to assess answer reliability, OneChart achieves more precise chart parsing than other models. Despite these efforts, existing chart parsing models still struggle with complex, high-density charts, frequently producing missing data, hallucinated data, or severe numerical deviations. Moreover, some of these models can offer the confidence score of their own answer but lack the ability to refine or self-correct incorrect outputs, which limits their applicability in high-accuracy scenarios.

## D   CASES FROM REAL-WORLD CHARTS IN CHARTP-BENCH

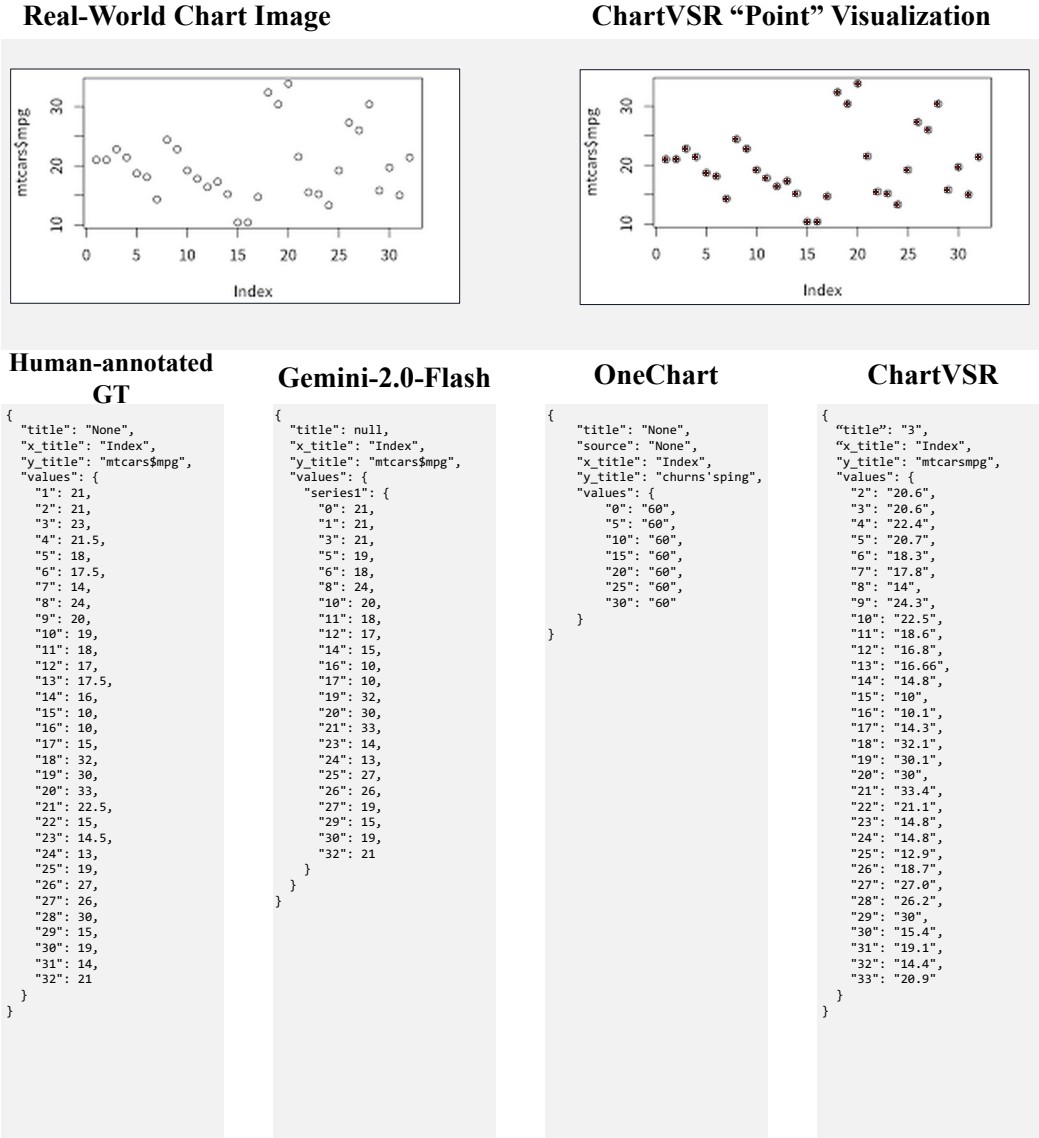

Figure 7: **case 1**

**Real-World Chart Image**

**ChartVSR "Point" Visualization**

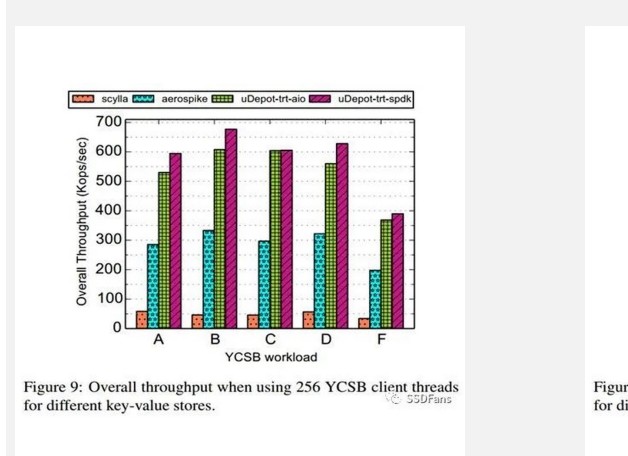

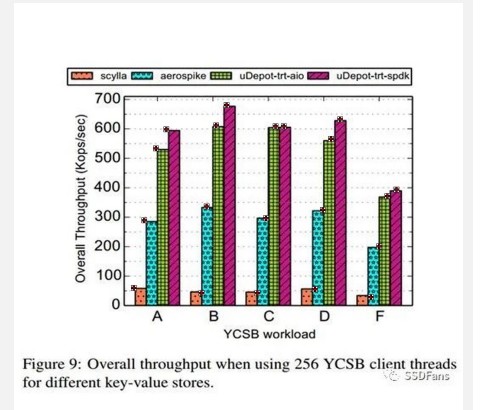

Figure 9: Overall throughput when using 256 YCSB client threads for different key-value stores.

Figure 9: Overall throughput when using 256 YCSB client threads for different key-value stores.

**Human-annotated GT**

```json
{
 "title": "None",
 "x_title": "YCSB workload",
 "y_title": "Overall
Throughput (Kops/sec)",
 "values": {
  "scylla": {
   "A": 55,
   "B": 45,
   "C": 45,
   "D": 53,
   "F": 40
  },
  "aerospike": {
   "A": 280,
   "B": 330,
   "C": 300,
   "D": 320,
   "F": 200
  },
  "uDepot-trt-aio": {
   "A": 530,
   "B": 610,
   "C": 605,
   "D": 560,
   "F": 370
  },
  "uDepot-trt-spdk": {
   "A": 595,
   "B": 680,
   "C": 605,
   "D": 630,
   "F": 390
  }
 }
}
```

**Gemini-2.0-Flash**

```json
{
 "title": "Overall
throughput when using 256
YCSB client threads for
different key-value stores.",
 "x_title": "YCSB workload",
 "y_title": "Overall
Throughput (Kops/sec)",
 "values": {
  "scylla": {
   "A": 60,
   "B": 40,
   "C": 40,
   "D": 50,
   "F": 30
  },
  "aerospike": {
   "A": 280,
   "B": 330,
   "C": 290,
   "D": 310,
   "F": 200
  },
  "uDepot-trt-aio": {
   "A": 530,
   "B": 600,
   "C": 600,
   "D": 560,
   "F": 370
  },
  "uDepot-trt-spdk": {
   "A": 590,
   "B": 670,
   "C": 600,
   "D": 620,
   "F": 390
  }
 }
}
```

**OneChart**

```json
{
 "title": "None",
 "source": "None",
 "x_title": "None",
 "y_title": "Overall
Throughput (Kosp/SEC)",
 "values": {
  "scycla": {
   "A": "28.5",
   "B": "32.1",
   "C": "29.8",
   "C": "31.2",
   "C": "34.6",
   "C": "39.8",
   "C": "42.4",
   "C": "48.2",
   "C": "50.5"
  },
  "Aerospace": {
   "A": "35.2",
   "B": "33.8",
   "C": "40.5",
   "C": "45.2",
   "C": "43.9",
   "C": "45.6",
   "C": "47.8",
   "C": "48.9",
   "C": "52.1"
  },
  "Other-tratt-spdK": {
   "A": "60.3",
   "B": "58.9",
   "C": "60.1",
   "C": "58.7",
   "C": "62.9",
   "C": "59.4",
   "C": "53.8",
   "C": "53.6",
   "C": "53.2",
   "C": "53.8"
  }
 }
}
```

**ChartVSR**

```json
{
 "title": "figure 9",
 "x_title": "YSCB
workload",
 "y_title": "Overall
throughput (Kops/sec)",
 "values": {
  "scylla": {
   "A": "54.9",
   "B": "41.0",
   "C": "41.0",
   "D": "53.2",
   "F": "38.3"
  },
  "aerospike": {
   "A": "280",
   "B": "325.7",
   "C": "287",
   "D": "314.1",
   "F": "194.0"
  },
  "uDepot trt-aio": {
   "A": "529.0",
   "B": "610",
   "C": "607.0",
   "D": "560",
   "F": "360"
  },
  "uDepot trt-spdk": {
   "A": "596.9",
   "B": "678.0",
   "C": "607.0",
   "D": "630",
   "F": "382"
  }
 }
}
```

Figure 8: **case 2**

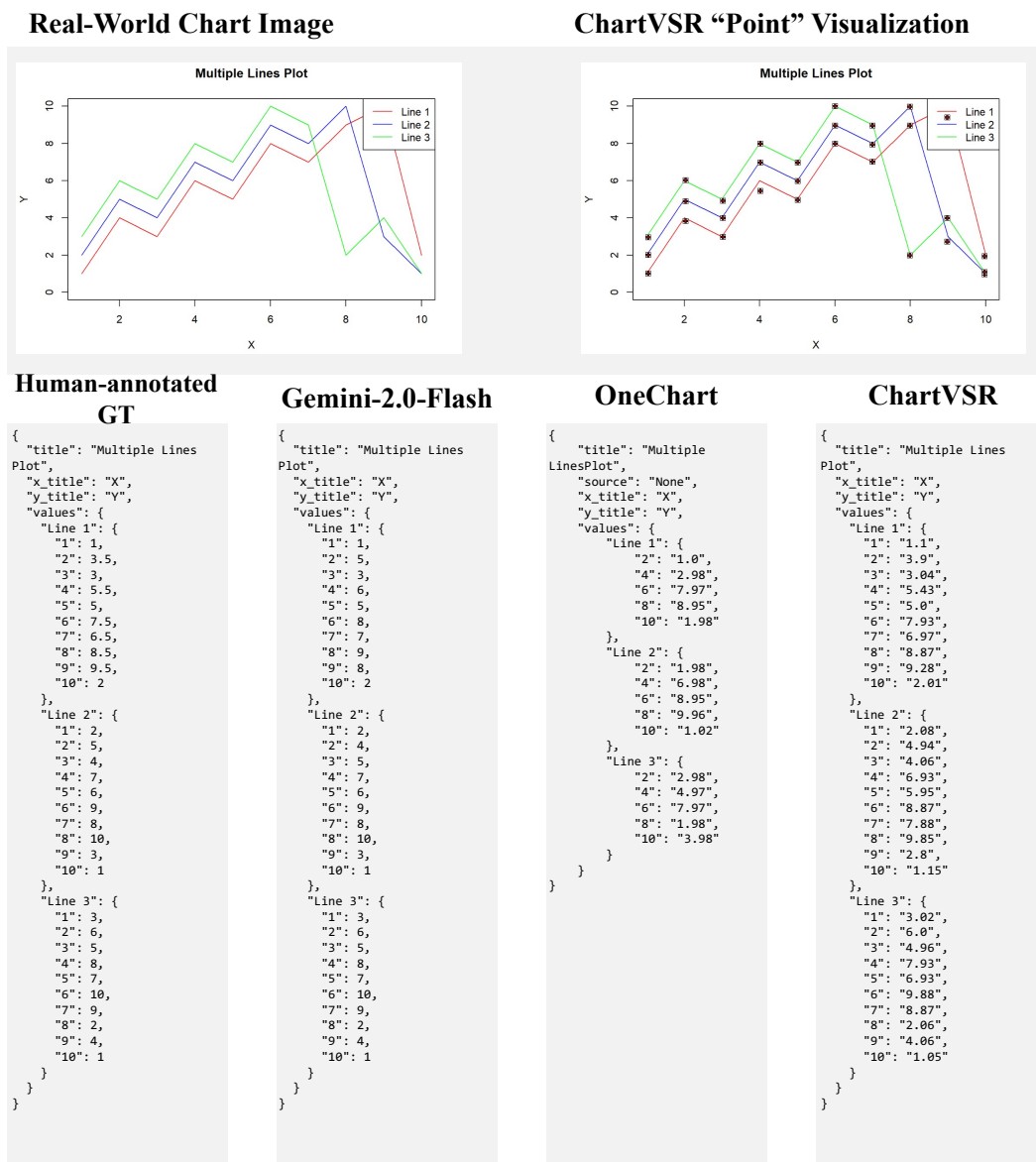

Figure 9: **case 3**

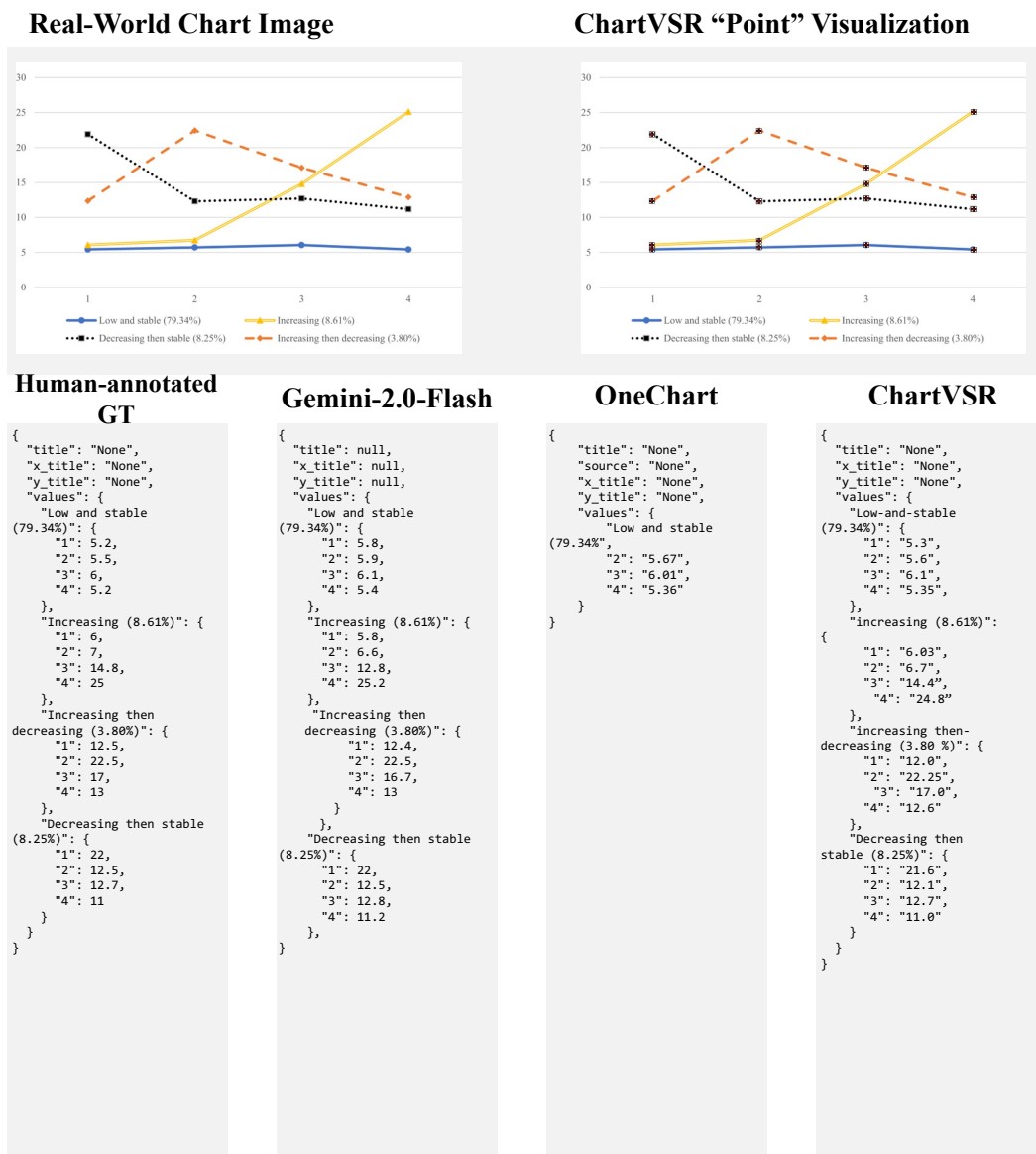

**Real-World Chart Image**

**ChartVSR "Point" Visualization**

**Human-annotated GT**

```
{
  "title": "None",
  "x_title": "None",
  "y_title": "None",
  "values": {
    "Low and stable
(79.34%)": {
      "1": 5.2,
      "2": 5.5,
      "3": 6,
      "4": 5.2
    },
    "Increasing (8.61%)": {
      "1": 6,
      "2": 7,
      "3": 14.8,
      "4": 25
    },
    "Increasing then
decreasing (3.80%)": {
      "1": 12.5,
      "2": 22.5,
      "3": 17,
      "4": 13
    },
    "Decreasing then stable
(8.25%)": {
      "1": 22,
      "2": 12.5,
      "3": 12.7,
      "4": 11
    }
  }
}
```

**Gemini-2.0-Flash**

```
{
  "title": null,
  "x_title": null,
  "y_title": null,
  "values": {
    "Low and stable
(79.34%)": {
      "1": 5.8,
      "2": 5.9,
      "3": 6.1,
      "4": 5.4
    },
    "Increasing (8.61%)": {
      "1": 5.8,
      "2": 6.6,
      "3": 12.8,
      "4": 25.2
    },
    "Increasing then
decreasing (3.80%)": {
      "1": 12.4,
      "2": 22.5,
      "3": 16.7,
      "4": 13
    },
    "Decreasing then stable
(8.25%)": {
      "1": 22,
      "2": 12.5,
      "3": 12.8,
      "4": 11.2
    },
  }
}
```

**OneChart**

```
{
  "title": "None",
  "source": "None",
  "x_title": "None",
  "y_title": "None",
  "values": {
    "Low and stable
(79.34%",
      "2": "5.67",
      "3": "6.01",
      "4": "5.36"
  }
}
```

**ChartVSR**

```
{
  "title": "None",
  "x_title": "None",
  "y_title": "None",
  "values": {
    "Low-and-stable
(79.34%)": {
      "1": "5.3",
      "2": "5.6",
      "3": "6.1",
      "4": "5.35",
    },
    "increasing (8.61%)":
{
      "1": "6.03",
      "2": "6.7",
      "3": "14.4",
      "4": "24.8"
    },
    "increasing then-
decreasing (3.80 %)": {
      "1": "12.0",
      "2": "22.25",
      "3": "17.0",
      "4": "12.6"
    },
    "Decreasing then
stable (8.25%)": {
      "1": "21.6",
      "2": "12.1",
      "3": "12.7",
      "4": "11.0"
    }
  }
}
```

Figure 10: **case 4**

**Real-World Chart Image**

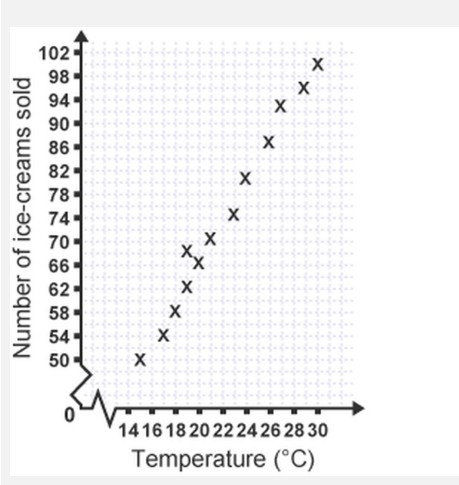

**ChartVSR "Point" Visualization**

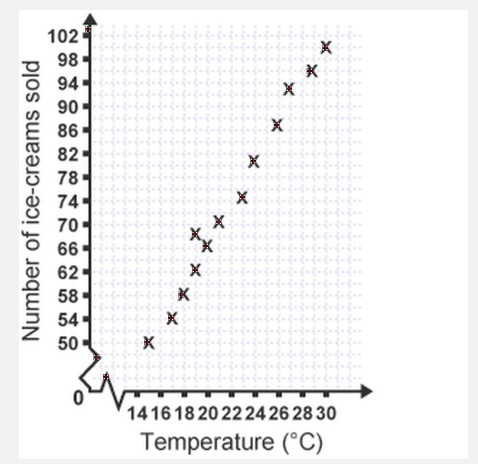

**Human-annotated GT**

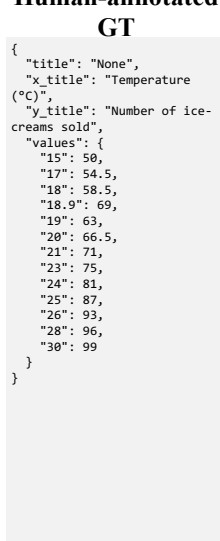

```
{
  "title": "None",
  "x_title": "Temperature
(°C)",
  "y_title": "Number of ice-
creams sold",
  "values": {
    "15": 50,
    "17": 54.5,
    "18": 58.5,
    "18.9": 69,
    "19": 63,
    "20": 66.5,
    "21": 71,
    "23": 75,
    "24": 81,
    "25": 87,
    "26": 93,
    "28": 96,
    "30": 99
  }
}
```

**Gemini-2.0-Flash**

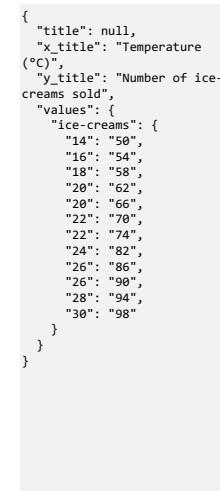

```
{
  "title": null,
  "x_title": "Temperature
(°C)",
  "y_title": "Number of ice-
creams sold",
  "values": {
    "ice-creams": {
      "14": "50",
      "16": "54",
      "18": "58",
      "20": "62",
      "20": "66",
      "22": "70",
      "22": "74",
      "24": "82",
      "26": "86",
      "26": "90",
      "28": "94",
      "30": "98"
    }
  }
}
```

**OneChart**

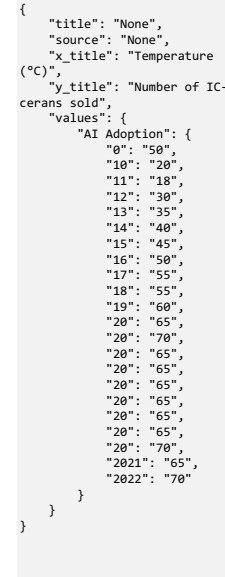

```
{
  "title": "None",
  "source": "None",
  "x_title": "Temperature
(°C)",
  "y_title": "Number of IC-
cerans sold",
  "values": {
    "AI Adoption": {
      "0": "50",
      "10": "20",
      "11": "18",
      "12": "30",
      "13": "35",
      "14": "40",
      "15": "45",
      "16": "50",
      "17": "55",
      "18": "55",
      "19": "60",
      "20": "65",
      "20": "70",
      "20": "65",
      "20": "65",
      "20": "65",
      "20": "65",
      "20": "65",
      "20": "70",
      "2021": "65",
      "2022": "70"
    }
  }
}
```

**ChartVSR**

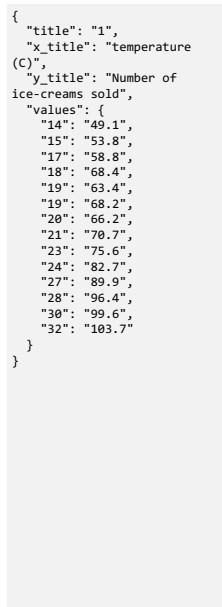

```
{
  "title": "1",
  "x_title": "temperature
(C)",
  "y_title": "Number of
ice-creams sold",
  "values": {
    "14": "49.1",
    "15": "53.8",
    "17": "58.8",
    "18": "68.4",
    "19": "63.4",
    "19": "68.2",
    "20": "66.2",
    "21": "70.7",
    "23": "75.6",
    "24": "82.7",
    "27": "89.9",
    "28": "96.4",
    "30": "99.6",
    "32": "103.7"
  }
}
```

Figure 11: **case 5**

# E  EXPLANATIONS FOR DATASET COMPARISON METRIC

**Average Data Points per Chart** (↑)   This metric measures the average visual complexity and information density of the charts. A higher value indicates that the dataset contains more intricate charts with a larger number of data elements, which poses a greater challenge for chart parsing models and is more representative of complex real-world visualizations.

**Unique Numerical Ratio** (↑)   This ratio is calculated as the number of unique numerical values divided by the total number of numerical values across the entire dataset. It serves as a direct measure of data value diversity. A low ratio suggests high data homogeneity and value repetition, which can cause models to overfit to specific numerical patterns. Our dataset's high ratio demonstrates a significant diversity in the underlying data values.

**Average Absolute Correlation Coefficient** (↓)   To measure the diversity of data trends, we calculate the Pearson correlation coefficient for all multi-series charts and average their absolute values. A score close to 1.0 indicates that the dataset is dominated by monotonic trends (strong positive or negative correlations), a common issue in synthesized datasets. A lower score, as achieved by our dataset, signifies a richer variety of data relationships, including weak correlations, non-linear patterns, and random distributions, thus preventing models from developing biases toward simple linear trends.

**Average PMI of Top-K Label Pairs** (↓)   This metric quantifies the prevalence of "implicit regularities" by measuring the Pointwise Mutual Information (PMI) between the most frequent label pairs. A high PMI score reveals that certain labels (e.g., '2020' and '2021', or 'USA' and 'China') co-occur far more frequently than by chance, creating predictable patterns. While our dataset's PMI is competitive and avoids the extreme regularities seen in ChartX, ChartQA exhibits the lowest PMI, indicating the least predictable label associations among the compared datasets.

**User Study**   To assess the perceived visual diversity and realism of the generated charts, we conducted a comprehensive user study with 14 volunteers. For each of the four datasets (Ours, ChartQA, PlotQA, and ChartX), we randomly selected and compiled 16 chart images into a single 4x4 grid. In each of the 50 trials presented to a volunteer, they were shown the four grids side-by-side. The volunteers were then asked to rate the overall stylistic diversity and visual quality of each grid on a **4-point Likert scale** (1: Poor, 2: Fair, 3: Good, 4: Excellent). The order of the grids was randomized in each trial to prevent positional bias. The results, summarized in Table 1, show that our dataset was rated significantly higher, confirming its superior visual variety and quality from a human perspective.

# F  STRUCTURING CHART-ORIENTED REPRESENTATION METRIC

We adopt SCRM (Structuring Chart-oriented Representation Metric) (Xia et al., 2023) as the evaluation metric. SCRM quantifies the structural consistency between a model's chart–parsing result and the ground truth at two levels, image and dataset. Given the $p$-th predicted triplet and the $q$-th ground-truth (GT) triplet, **entity similarity** $J(p, q)$ is measured by the edit distance, and **value error** $e(p, q)$ is measured by the relative difference. A predicted/GT pair is regarded as matched when the two conditions $J(p, q) \leq J_{thr}$ and $e(p, q) \leq e_{thr}$ are both satisfied:

$$\ell(p, q) = \begin{cases} 1, & \text{if } J(p, q) \leq J_{thr} \ \wedge \ e(p, q) \leq E_{thr} \\ 0, & \text{otherwise} \end{cases} \tag{1}$$

Let $P$ and $Q$ be the numbers of predicted and GT triplets in the image, respectively. The Image-level structural IoU is

$$\text{IoU} = \frac{\sum_{p=1}^{P} \sum_{q=1}^{Q} \ell(p, q)}{P + Q - \sum_{p=1}^{P} \sum_{q=1}^{Q} \ell(p, q)} \tag{2}$$

Then, for the $i$-th chart in a dataset of $L$ charts, define

$$d(i, t) = \begin{cases} 1, & \text{if IoU}(i) \geq t \\ 0, & \text{otherwise} \end{cases} \tag{3}$$

The Dataset-level precision is then:

$$\text{AP}(IoU_{thr}) \ = \ \frac{1}{|IoU_{thr}| \, L} \sum_{t \in IoU_{thr}} \sum_{i=1}^{L} d(i, t) \tag{4}$$

In our experiments, we use the standard SCRM, which adopts the threshold set $IoU_{\text{thr}} = 0.50, 0.55, \ldots, 0.95$.

## G   ERRONEOUS SAMPLES IN CHARTQA-SE

As shown in Figure 12 and Figure 13, we present some representative error samples in ChartQA-SE. The complete list of erroneous samples is provided in the "chartqa_se_wrong.json".

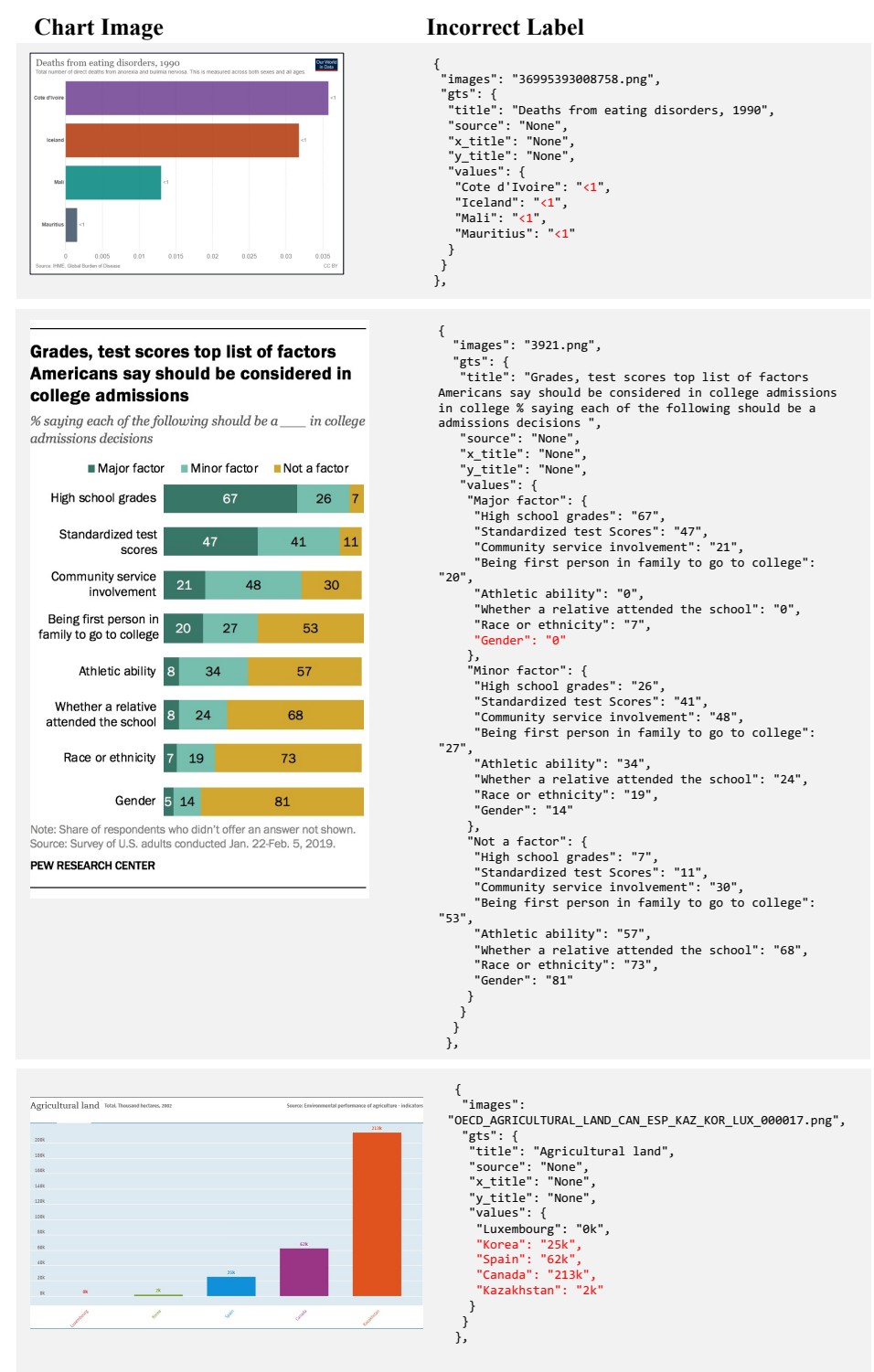

Figure 12: **Erroneous samples in ChartQA-SE**

**Chart Image**                                                    **Incorrect Label**

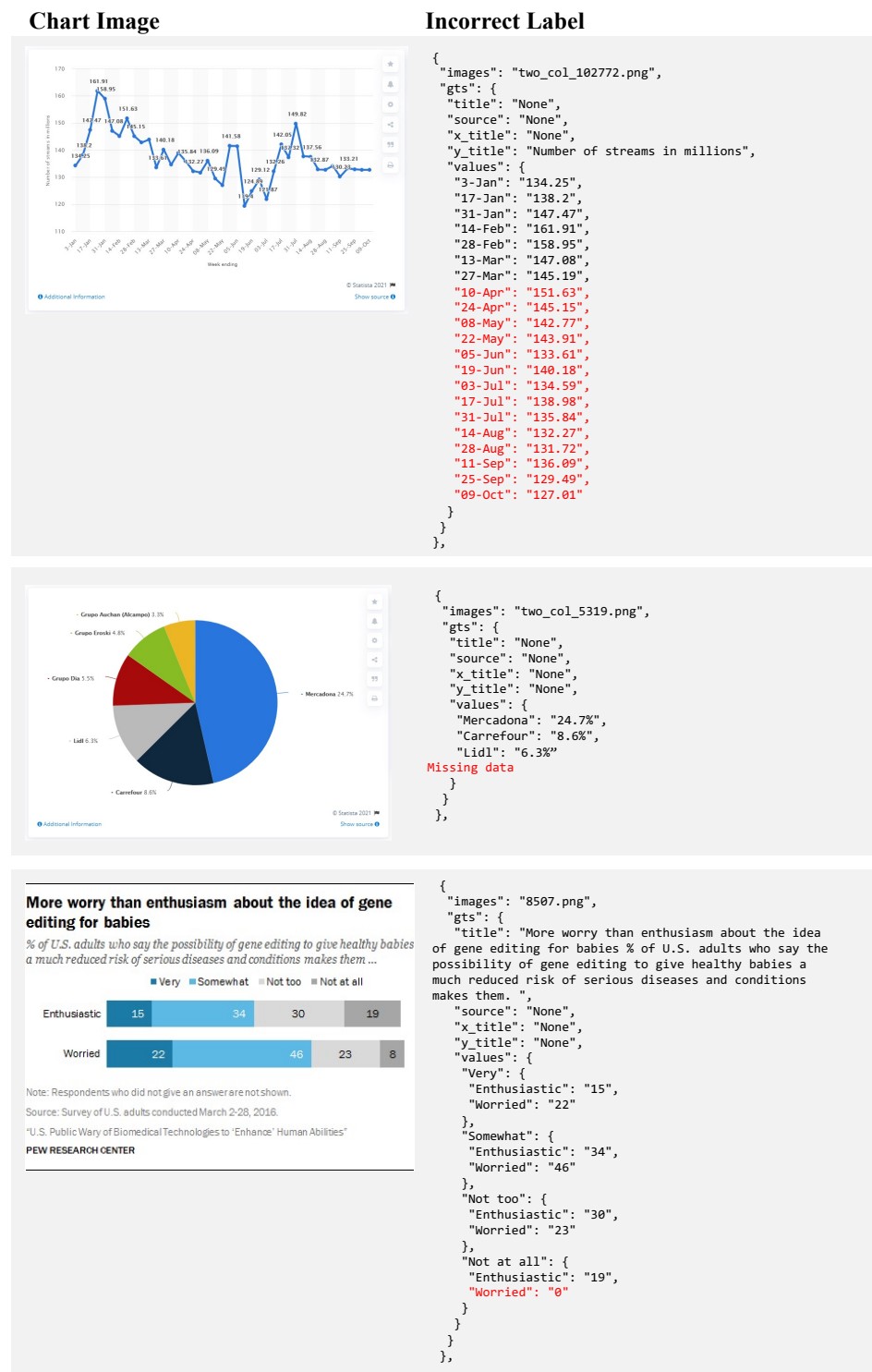

Figure 13: **Erroneous samples in ChartQA-SE**

# H STUBBORN ERRORS AFTER THE FIRST REFINEMENT ROUND

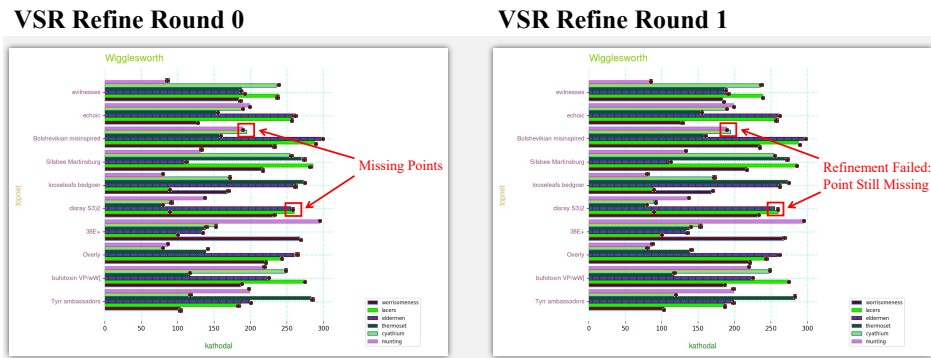

Figure 14: **High Visual Density** In charts with extreme data density, the model struggles to distinguish individual elements within the dense clusters, leading to persistent verification failures.

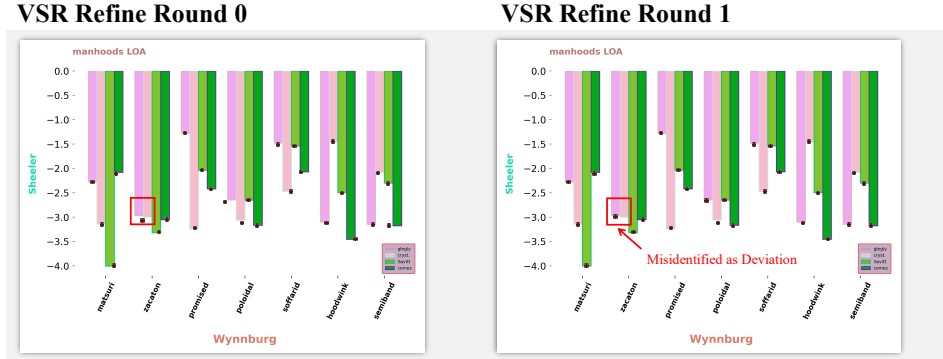

Figure 15: **Error Misinterpretation and Shifting** The model sometimes misidentifies the type of error, causing a "correction" that shifts the mistake rather than solving it. For instance, a missing point might be misinterpreted as a large deviation of a nearby point. The model then "corrects" this by moving the existing point to the missing location, effectively fixing the omission but creating a new vacancy where the original point stood.

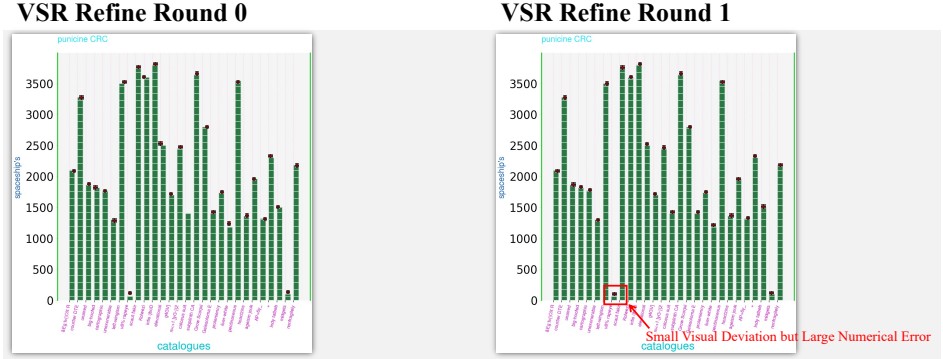

Figure 16: **Visual Insensitivity to Relative Precision** This typically occurs in charts with large data disparities (e.g., a range containing both 10,000 and 10). A deviation of merely a few pixels is visually imperceptible in the context of the large scale. However, for the small data values (e.g., 10), this tiny pixel shift translates to a massive error (e.g., 50%), causing the model to incorrectly confirm the visual position as "accurate" while it fails the numerical evaluation metric.

