# OpenReview forum: "Visual Self-Refine: A Pixel-Guided Paradigm for Accurate Chart Parsing"
_ICLR.cc/2026/Conference — ICLR 2026 Poster_

### Official Review · Reviewer_tCHt · 2025-10-23

**Soundness:** 3
**Presentation:** 3
**Contribution:** 3
**Rating:** 6
**Confidence:** 4

**Summary:**

This paper introduces Visual Self-Refine (VSR), a novel paradigm to address the poor performance of LVLMs on vision-centric tasks like chart parsing, where text-based self-correction fails. Inspired by the human strategy of using a finger as a "visual anchor" , VSR enables a model to generate pixel-level localizations, visualize them on the image, and then feed this visualization back to itself to iteratively inspect and correct visual perception errors like omissions or hallucinations. The authors instantiate this paradigm in a model called ChartVSR, which uses a Refine Stage to verify pixel locations and a Decode Stage to parse the chart using these verified anchors. This work is supported by two additional contributions: a new challenging ChartP-Bench benchmark and a robust data engine for generating a large-scale , diverse training dataset.

**Strengths:**

- The paper is very well-written and very clear.
- The paper conducted abundant analysis.
- Meaningful contributions in both data and method.
- The potential applications to other precision-oriented tasks like Visual Counting and Visual Grounding are clearly articulated (Figure 6) , opening a promising new research direction.

**Weaknesses:**

- In Table 2 and 3, the authors are comparing open-source models of different sizes. The author should clearly list the sizes of these models for fairer comparison. Additionally, to better show the effectiveness of the proposed training data, the author can consider training the MatCha model with their data and do direct comparison with DePlot.
- For chart parsing evaluation, the paper should also compare with the Chart-to-Text [2] and AskChart [3].




[1] MatCha: Enhancing Visual Language Pretraining with Math Reasoning and Chart Derendering. ACL 2023

[2] Do LVLMs Understand Charts? Analyzing and Correcting Factual Errors in Chart Captioning. ACL 2024 Findings

[3] AskChart: Universal Chart Understanding through Textual Enhancement.

**Questions:**

Could you provide more qualitative or quantitative analysis on the "stubborn errors" that VSR fails to correct after the first round? Are they of a specific type (e.g., heavily occluded points, ambiguous labels, tiny markers)?

---

> ### Author Response · Authors · 2025-11-26
>
> We sincerely thank Reviewer tCHt for the positive assessment and for highlighting the **clarity of our writing**, the **abundance of our analysis**, and the **meaningful contributions** of both our VSR paradigm and the ChartP-Bench. We are especially glad that Reviewer tCHt recognizes the potential of VSR to open promising new research directions in other precision-oriented tasks.
>
> ---
>
> **Weakness 1: In Table 2 and 3, the authors are comparing open-source models of different sizes. The author should clearly list the sizes of these models for fairer comparison. Additionally, to better show the effectiveness of the proposed training data, the author can consider training the MatCha model with their data and do direct comparison with DePlot.**
>
> **Response 1:** We thank Reviewer tCHt for this valuable suggestion to improve the fairness and transparency of our model comparisons.
>
> 1.  **Model Parameter Counts:** We have updated our experimental in Table 2 and Table 3 to explicitly list the parameter counts for all compared models.
>
> | Model | DePlot | ChartAst | ChartVLM | OneChart | ChartVSR |
> | :--- | :---: | :---: | :---: | :---: | :---: |
> | **Size** | 1.3B | 13B | 7.3B | 0.2B | **3B** |
>
> 2.  **MatCha Training:** We fully agree that training a MatCha model on our proposed data would be an excellent ablation study to isolate the specific contribution of our Data Engine. However, MatCha is based on the Pix2Struct architecture, which differs significantly from modern LVLMs, and training it from scratch demands substantial computational resources. We are currently in the process of applying for the necessary computational resources to conduct this experiment. Once the experiment is completed, we intend to include these results. In the interim, we believe the comparison with **DePlot**, which is essentially a MatCha model fine-tuned on chart-to-table tasks, serves as a robust baseline to represent the capabilities of that model family.
>
> ---
>
> **Weakness 2: For chart parsing evaluation, the paper should also compare with the Chart-to-Text [2] and AskChart [3].**
>
> **Response 2:** We thank Reviewer tCHt for pointing out these relevant works. We have attempted to evaluate both methods on our **ChartP-Bench** to provide a more comprehensive comparison.
>
> Regarding **AskChart [3]**, we were unfortunately unable to conduct the evaluation because the official model weights have not yet been released (the official repository states that the model weights are still on the "to-do list").
>
> Regarding the **[2]** baseline, we evaluated it on ChartP-Bench and the results are presented in the table below. The model achieves near-zero performance on the SCRM metrics.
>
> | Model | Easy-Strict | Easy-Slight | Easy-High | Hard-Strict | Hard-Slight | Hard-High |
> | :--- | :---: | :---: | :---: | :---: | :---: | :---: |
> | **Chart-To-Table [2]** | **0.00** | **0.00** | **0.05** | **0.00** | **0.00** | **0.00** |
> | Deplot | 0.00 | 9.73 | 15.35 | 0.00 | 1.27 | 4.19 |
> | OneChart | 0.00 | 4.75 | 18.89 | 0.00 | 1.98 | 9.48 |
> | Gemini-2.5-Pro | 0.06 | 36.51 | 46.05 | 0.00 | 46.11 | 66.39 |
> | ChartVSR (Ours) | 0.11 | 51.95 | 67.44 | 0.02 | 49.30 | 63.67 |

---

> > ### Author Response · Authors · 2025-11-26
> >
> > ---
> >
> > **Question 1: Could you provide more qualitative or quantitative analysis on the "stubborn errors" that VSR fails to correct after the first round? Are they of a specific type?**
> >
> > **Response 3:** We appreciate this insightful question. We have conducted a deeper error analysis on the samples that remain incorrect after the first refinement round (the "stubborn errors") and have visualized these representative cases in **Appendix H**. As shown in **Figures 14, 15, and 16**, these errors primarily fall into three specific categories:
> >
> > 1.  **High Visual Density (Figure 14):** In charts with extreme data density, the model struggles to distinguish individual elements within the dense clusters, leading to persistent verification failures.
> >
> > 2.  **Error Misinterpretation and Shifting (Figure 15):** The model sometimes misidentifies the *type* of error, causing a "correction" that shifts the mistake rather than solving it. For instance, a **missing point** might be misinterpreted as a **large deviation** of a nearby point. The model then "corrects" this by moving the existing point to the missing location, effectively fixing the omission but creating a new vacancy where the original point stood.
> >
> > 3.  **Visual Insensitivity to Relative Precision (Figure 16):** This typically occurs in charts with large data disparities (e.g., a range containing both 10,000 and 10). A deviation of merely a few pixels is visually imperceptible in the context of the large scale. However, for the small data values (e.g., 10), this tiny pixel shift translates to a massive error (e.g., >30%), causing the model to incorrectly confirm the visual position as "accurate" while it fails the numerical evaluation metric.
> >
> > ---
> >
> > If you still have any concerns or aspects you would like to discuss further, please do not hesitate to contact us at any time.
> >
> > **We sincerely thank you for your time and thoughtful comments. If our response has addressed your concerns, we would deeply appreciate your consideration of raising your rating. We greatly value your feedback and, regardless, sincerely appreciate your engagement with our work.**
> >
> > Best regards,
> >
> > The Authors

---

> ### Comment · Reviewer_tCHt · 2025-11-27
>
> Your response has addressed my concerns. I have increased my rating.

---

### Official Review · Reviewer_jBBM · 2025-10-30

**Soundness:** 2
**Presentation:** 2
**Contribution:** 2
**Rating:** 4
**Confidence:** 5

**Summary:**

The authors first had the model generate pixel-level localization results, visualized these localizations and plotted them back onto the original image, then fed this labeled image back to the model, allowing the model to self-check and correct visual perception errors as if "checking point by point with a finger," and finally performed structured parsing based on the confirmed pixel anchors.

**Strengths:**

A visual self-refinement paradigm, Visual Self-Refine (VSR), is proposed: the model first generates localization points, visualizes them, and then feeds them back to the model for self-checking and error correction.

In the graph parsing task, the process is divided into two stages: Refine and Decode.

A challenging benchmark, ChartP-Bench, is constructed, and ChartQA-SE is cleaned to obtain ChartQA-SE-Clean. Significant performance is reported on multiple benchmarks, especially outperforming strongly closed-source models (such as Gemini-2.5-Pro) and existing dedicated models on ChartP-Bench.

**Weaknesses:**

There are limited benchmarks for evaluating papers, lacking authoritative datasets like Chart-Pro and ChartXiv.

This method has limited nooverty, and its two-stage design is very similar to the design philosophy of SoM. Many previous works on visual prompts have demonstrated that such visual prompts can improve performance.

It lacks a crucial baseline, such as a comparison of the localization ability of the first-stage model with that of other grounding models on chart localization tasks.

**Questions:**

Will the model trained after refine + decode experience a performance degradation on refinement tasks? Quantitative results are needed.


There are limited benchmarks for evaluating papers, lacking authoritative datasets like Chart-Pro and ChartXiv.

---

> ### Author Response · Authors · 2025-11-26
>
> > We sincerely appreciate Reviewer jBBM for the time and effort dedicated to reviewing our work. **We notice that several key concerns, specifically those regarding benchmark selection and baseline comparisons, might stem from a potential misunderstanding of the specific domain and operational mechanism of our proposed method.** We would like to respectfully clarify that our work focuses strictly on **Chart Parsing** rather than **Chart QA**. Furthermore, our **VSR paradigm** serves as a **generative self-correction mechanism** rather than a standard **visual grounding** task. We hope the following detailed responses will help clarify these critical distinctions and address these concerns effectively.
>
> We sincerely thank Reviewer jBBM for the constructive feedback and for recognizing the value of our work. We are particularly encouraged that the reviewer appreciates the **proposed Visual Self-Refine (VSR) paradigm**, the **logical two-stage design (Refine and Decode)**, and the **construction of the challenging ChartP-Bench**, where our model achieves **significant performance superior to strong closed-source models**.
>
> ---
>
> **Weakness 1 & Question 2: There are limited benchmarks for evaluating papers, lacking authoritative datasets like Chart-Pro and ChartXiv.**
>
> **Response 1:** We respectfully clarify that **Chart Parsing** and **Chart Question Answering (QA)** are fundamentally different tasks with distinct evaluation goals. Datasets like Chart-Pro and ChartXiv are indeed authoritative, but they are designed for high-level reasoning and QA, requiring natural language answers rather than precise structural data recovery. Chart Parsing aims to extract the underlying data table (exact coordinates and values) from the image, which serves as a foundational step for downstream tasks. Consequently, QA benchmarks lack the necessary structural ground truth (such as JSON or data tables) to evaluate the precision of a parsing model. We focused our evaluation on **ChartQA-SE, PlotQA-SE, ChartX-SE, and our ChartP-Bench** because they provide the specific structured annotations required to rigorously assess chart parsing accuracy.
>
> ---
>
> **Weakness 2: This method has limited novelty, and its two-stage design is very similar to the design philosophy of SoM.**
>
> **Response 2:** We must respectfully clarify a fundamental distinction between our work and Set-of-Mark (SoM), as they represent **completely different paradigms**. SoM is a **visual prompting** technique that relies on **external** segmentation models (like SAM) to generate static, pre-defined masks *before* inference to help the model "ground" its attention. It essentially says, "Look at these specific regions I marked for you." In stark contrast, VSR is a **visual self-correction** paradigm. Our model generates its **own** pixel-level hypotheses from scratch, visualizes them, and then feeds this visualization back to itself to verify correctness. This is a dynamic **"generate-evaluate-refine" loop** designed to enable visual reasoning and error correction, mimicking human introspection. While SoM provides a static map for guidance, VSR builds an autonomous agent capable of reflecting on its own outputs. Equating VSR with SoM overlooks this core contribution: VSR is not about how to prompt a model, but how to enable a model to visualize its own thoughts to fix its own mistakes.

---

> > ### Author Response · Authors · 2025-11-26
> >
> > ---
> >
> > **Weakness 3: It lacks a crucial baseline, such as a comparison of the localization ability of the first-stage model with that of other grounding models on chart localization tasks.**
> >
> > **Response 3:** We respectfully clarify the fundamental difference between the **Chart Layout Grounding** task mentioned by the reviewer and the task performed in our Refine Stage.
> >
> > 1.  **Fundamental Task Difference:** Standard chart localization or grounding models typically perform **Layout Detection**, outputting **Bounding Boxes** to locate semantic regions (e.g., "plot area", "legend", "axis title"). In contrast, our Refine Stage performs **Pixel-level Visual Anchor Estimation**, outputting specific **(x, y) coordinates** that represent the data values (e.g., the exact top pixel of a bar or the vertex of a line).
> > 2.  **Infeasibility of Direct Comparison:** It is technically infeasible to quantitatively compare "Bounding Boxes" against "Visual Anchor." A bounding box covers an entire object, whereas a visual anchor a precise value reading location. Metrics like IoU applicable to boxes cannot be applied to single pixels.
> > 3.  **Different Objectives:** The goal of standard grounding is perception (finding where elements are). The goal of our localization is **Visual Self-Refine**, visualizing the model's *own* intermediate results to enable self-correction. Comparing our internal reflection mechanism against an external detection model is not an apples-to-apples comparison.
> >
> > To our knowledge, there are no existing baselines that perform this specific **generative pixel-level self-refinement** on charts. However, if the reviewer is aware of specific prior works that perform this exact point-based self-correction task, we would strictly welcome the references and include them for discussion.
> >
> > ---
> >
> > **Question 1: Will the model trained after refine + decode experience a performance degradation on refinement tasks? Quantitative results are needed.**
> >
> > **Response 4:** We thank the reviewer for this insightful question. First, it is important to clarify the distinct roles of the two stages: the **Refine Stage** operates in **Pixel Space** to generate precise visual anchors, while the **Decode Stage** maps these anchors to **Chart Space**. Without the Decode stage, the model cannot produce the final structured chart data. Regarding the concern about potential interference, we conducted an experiment comparing a model trained solely on the Refine task against our final ChartVSR model (trained on both Refine and Decode). As shown in the table below, the **Localization Accuracy** remains stable and high. This suggests that the two tasks are synergistic rather than conflicting. The Decode stage introduces high-level semantic understanding (e.g., understanding that a series should be continuous across the x-axis or that specific legends correspond to distinct data streams). This semantic context reinforces the localization process. For instance, even if two lines overlap at a specific point making visual detection difficult, the semantic expectation learned from the Decode task, that a data point should exist there for that category, helps the model effectively prevent omissions and maintain high localization recall.
> >
> > | Model | Localization Accuracy (Easy Subset) | Localization Accuracy (Hard Subset) |
> > | :--- | :---: | :---: |
> > | Refine Only | 90.9% | 84.5% |
> > | Refine + Decode | 91.7% | 86.1% |
> >
> > ---
> >
> > If you still have any concerns or aspects you would like to discuss further, please do not hesitate to contact us at any time.
> >
> > **We sincerely thank you for your time and thoughtful comments. If our response has addressed your concerns, we would deeply appreciate your consideration of raising your rating. We greatly value your feedback and, regardless, sincerely appreciate your engagement with our work.**
> >
> > Best regards,
> >
> > The Authors

---

> > > ### Comment · Reviewer_jBBM · 2025-11-28
> > >
> > > The author's reply resolved my question, so I chose to increase my score.

---

### Official Review · Reviewer_c4Zx · 2025-10-30

**Soundness:** 2
**Presentation:** 3
**Contribution:** 3
**Rating:** 6
**Confidence:** 3

**Summary:**

This paper proposes Visual Self-Refine (VSR), a novel paradigm that introduces visual feedback as a self-correction mechanism for LVLMs in visually intensive tasks such as chart parsing. The method, instantiated as ChartVSR, decomposes the parsing process into two stages: the Refine stage and the Decode stage. The authors also present ChartP-Bench, a new and challenging benchmark featuring visually dense and stylistically diverse charts. Experimental results demonstrate consistent improvements over both chart-specific and general-purpose LVLMs.

**Strengths:**

1. The paper presents a novel and interesting paradigm for chart parsing, introducing a visually grounded self-correction mechanism that enhances interpretability and addresses an existing gap in LVLM perception.
2. The authors introduce a high-quality dataset, ChartP-Bench, which is carefully curated, diverse in style, and fills an important gap in chart parsing evaluation.
3. The ablation studies are comprehensive, providing thorough analyses of the effects of pixel localization and refinement, and consistently demonstrating the benefits of the proposed approach.

**Weaknesses:**

1. The chart parsing paradigm proposed in this paper can be viewed as a type of reasoning paradigm. However, the experimental section lacks comparisons with other recent visual reasoning models, such as o1, Qwen3-VL, and InternVL-3.5. I understand that some of these models might not have been publicly available at the time of submission, but I recommend that the authors include such comparisons in future revisions to strengthen the solidity and comprehensiveness of the work.
2. Around line 405, the paper explains why the AP-Strict scores are so low. According to the authors, this metric only rewards models that output exactly correct numerical values. I find this requirement excessively strict, as it is practically impossible to infer such highly precise numbers (sometimes up to two or three decimal places) from a single image. Therefore, I have some reservations about the practical significance and interpretability of this metric.

**Questions:**

1. I am curious about the scalability and generalization ability of the proposed approach, particularly regarding its performance on unseen chart types and the relationship between the number of localized regions and the extent of performance improvement.
2. The open-source models achieve performance comparable to ChartVSR on existing datasets such as ChartQA-SE-Clean, but their performance drops significantly on the newly proposed ChartP-Bench. Could the authors clarify the reason for this discrepancy? Is it possible that the training data distribution of ChartVSR is similar to that of ChartP-Bench, giving it an advantage? If this is the case, I would suggest that the authors conduct an additional experiment in which the training set explicitly excludes data similar to ChartP-Bench in order to more convincingly demonstrate the model's out-of-domain generalization ability.
3. How does the proposed model perform on chart-related question-answering tasks such as ChartQA and PlotQA?

---

> ### Author Response · Authors · 2025-11-26
>
> We sincerely thank Reviewer c4Zx for the constructive comments and for recognizing the **novelty of our VSR paradigm**, the **high quality of our ChartP-Bench**, and the **comprehensiveness of our ablation studies**. We appreciate the opportunity to strengthen our work by addressing the concerns regarding baselines and generalization.
>
> ---
>
> **Weakness 1: The chart parsing paradigm proposed in this paper can be viewed as a type of reasoning paradigm. However, the experimental section lacks comparisons with other recent visual reasoning models, such as o1, Qwen3-VL, and InternVL-3.5. I understand that some of these models might not have been publicly available at the time of submission, but I recommend that the authors include such comparisons in future revisions to strengthen the solidity and comprehensiveness of the work.**
>
> **Response 1:** We thank the reviewer for this valuable suggestion and acknowledge the rapid pace of development in this field. Regarding OpenAI o1, as it is a closed-source model with high API costs and strict rate limits, performing a large-scale evaluation on the full benchmark is challenging. To address the request for newer open-source baselines, we have evaluated the latest state-of-the-art models, **Qwen3-VL** and **InternVL-3.5**, on ChartP-Bench. As shown in the table below, even compared to these newer baselines, ChartVSR maintains superior performance, further validating the effectiveness of the VSR paradigm.
>
> | Model | Easy-Strict | Easy-Slight | Easy-High | Hard-Strict | Hard-Slight | Hard-High |
> | :--- | :---: | :---: | :---: | :---: | :---: | :---: |
> | Qwen3-VL-8B | 0.00 | 19.61 | 31.33 | 0.00 | 20.18 | 30.14 |
> | Qwen3-VL-32B | 0.00 | 25.64 | 37.00 | 0.01 | 28.32 | 42.59 |
> | InternVL-3.5-8B | 0.00 | 30.05 | 43.44 | 0.00 | 29.48 | 44.47 |
> | InternVL-3.5-30B | 0.00 | 30.27 | 44.14 | 0.00 | 32.48 | 47.74 |
> | Gemini-2.5-Pro | 0.06 | 36.51 | 46.05 | 0.00 | 46.11 | 66.39 |
> | **ChartVSR (Ours)** | 0.11 | 51.95 | 67.44 | 0.02 | 49.30 | 63.67 |
>
> ---
>
> **Weakness 2: Around line 405, the paper explains why the AP-Strict scores are so low. According to the authors, this metric only rewards models that output exactly correct numerical values. I find this requirement excessively strict, as it is practically impossible to infer such highly precise numbers (sometimes up to two or three decimal places) from a single image. Therefore, I have some reservations about the practical significance and interpretability of this metric.**
>
> **Response 2:** We fully agree with the reviewer's assessment. The AP-Strict metric, which requires exact numerical equality or 0% error, is indeed extremely harsh and practically impossible for pixel-based estimation from a single image. We included this metric solely for consistency with previous literature, such as the original SCRM paper and OneChart, to ensure a fair comparison with prior arts. However, as the reviewer correctly noted, **AP-Slight** and **AP-High** are much more meaningful indicators of a model's structural parsing ability and practical utility. **We have reported these values in our paper (Table 2 and Table 3), and our model consistently demonstrates an advantage over baselines on these practical metrics.**
>
> ---

---

> > ### Author Response · Authors · 2025-11-26
> >
> > ---
> >
> > **Question 1: I am curious about the scalability and generalization ability of the proposed approach, particularly regarding its performance on unseen chart types and the relationship between the number of localized regions and the extent of performance improvement.**
> >
> > **Response 3:** Regarding scalability and generalization, **ChartP-Bench itself is constructed from "in the wild" images and inherently contains diverse and unseen chart types**, serving as a robust testbed for this capability. For chart types that are radically different or highly artistic and absent from the training distribution, performance naturally declines; this is a universal challenge for all chart parsing models and typically requires training to adapt. The diversity within ChartP-Bench explicitly exposes these challenges, and our model's leading performance confirms its relative robustness.
> >
> > ---
> >
> > **Question 2: Could the authors clarify the reason for the discrepancy between performance on existing datasets vs. ChartP-Bench? Is it possible that the training data distribution of ChartVSR is similar to that of ChartP-Bench?**
> >
> > **Response 4:** The performance gap primarily stems from the fact that existing models are typically trained on the **corresponding training sets** of benchmarks like ChartQA-SE or PlotQA-SE. This allows them to achieve high scores (e.g., AP-Slight > 80) due to their familiarity with the specific data distributions. However, **ChartP-Bench consists of "in the wild" charts that are completely unseen** and significantly more diverse. As shown in the table below, even the strongest models, including ChartVSR and Gemini-2.5-Pro, experience a significant performance drop to an AP-Slight of approximately 40 on this challenging benchmark. This sharp decline precisely validates our analysis in **Section 3.1**, which points out that existing training datasets often suffer from **stylistic homogeneity** and **implicit regularities**, causing models to overfit to specific patterns rather than learning true generalized parsing. As we further demonstrated in **Appendix D**, existing models struggle significantly when facing real-world variations due to these data limitations. ChartVSR's superior performance in this zero-shot setting demonstrates that our data engine effectively addresses these issues, imparting robust parsing capabilities that generalize better to wild data.
> >
> > ---
> >
> > **Question 3: How does the proposed model perform on chart-related question-answering tasks such as ChartQA and PlotQA?**
> >
> > **Response 5:** **Chart Parsing** and **Chart Question Answering (QA)** are fundamentally different tasks. Chart Parsing focuses on the precise extraction of structural data, whereas QA focuses on high-level reasoning. **ChartVSR is a specialized model designed specifically for high-precision parsing**, not a general-purpose QA model. Therefore, evaluating it on QA benchmarks is not applicable, as it does not output natural language answers. We believe that accurate parsing serves as a critical and independent foundation for reliable downstream applications.
> >
> > ---
> >
> > If you still have any concerns or aspects you would like to discuss further, please do not hesitate to contact us at any time.
> >
> > We sincerely thank you for your time and thoughtful comments. If our response has addressed your concerns, we would deeply appreciate your consideration of raising your rating. We greatly value your feedback and, regardless, sincerely appreciate your engagement with our work.
> >
> > Best regards,
> >
> > The Authors

---

### Official Review · Reviewer_NmPr · 2025-10-30

**Soundness:** 2
**Presentation:** 3
**Contribution:** 2
**Rating:** 6
**Confidence:** 4

**Summary:**

This paper proposes a methodology to improve chart parsing by including an additional step of providing pixel-level annotations on the charts in order to help the model improve precision. The refine step involves generating the pixel locations which are then used to annotate the image. Both the image and the pixel values are fed as input to the model which then makes a prediction based on this additional information. Experiments involve evaluating on a number of existing datasets as well as a new dataset introduced in this work. VSR results in nice improvements in some settings but the results aren’t consistent on existing datasets.

**Strengths:**

The new dataset seems nice and useful for chart parsing evals.
VSR is an interesting recipe and focusing on pixel-level annotations is a nice instantiation of this setup for chart parsing.
Some of the improvements seem compelling, particularly the information dense charts.
If the extra calls are prohibitive for an inference pipeline, this recipe can probably be used to create distillation data.

**Weaknesses:**

While the recipe is interesting, it’s not very general and will probably become outdated for this task as models’ visual understanding improves over time.
It seems strange that performance doesn’t improve much after a step or two of refinement even though there’s so much headroom. Why is this? Maybe annotations should be adjusted or focused on incorrect ones? Or doing step-by-step correction is necessary? Either way, it seems like the feedback and the recipe need some…refinement.

**Questions:**

see above

---

> ### Author Response · Authors · 2025-11-26
>
> We sincerely thank Reviewer NmPr for the constructive feedback and positive assessment. We are encouraged that the reviewer recognizes the **utility and quality of our new ChartP-Bench**, finds our **VSR recipe "interesting" and the focus on pixel-level annotations a "nice instantiation,"** and acknowledges the **"compelling" improvements on information-dense charts**. We also appreciate the insight regarding the potential use of our method for creating distillation data.
>
> ---
>
> **Weakness 1: While the recipe is interesting, it’s not very general and will probably become outdated for this task as models’ visual understanding improves over time.**
>
> **Response 1:** We appreciate this thoughtful perspective. While we agree that the visual capabilities of base models will continue to improve, we view the VSR paradigm as a complementary mechanism rather than a temporary fix. Even as models become stronger, achieving perfect zero-shot accuracy on dense, pixel-precise tasks remains challenging due to the inherent complexity of visual perception. VSR introduces an explicit "check-and-refine" loop, analogous to the "System 2" reasoning process where models or humans deliberate to ensure reliability. This mechanism allows the model to spot and correct errors that might occur in a single pass, regardless of the base model's strength. Furthermore, we believe VSR holds potential for broader applications beyond chart parsing. As preliminarily demonstrated in **Section 4.5 and Figure 6**, we applied VSR to tasks like **Visual Counting and Visual Grounding**, where the model visualizes intermediate outputs to self-verify. This suggests that the core idea of using visual feedback for self-correction can be adapted to various vision-centric tasks to enhance precision.
>
> ---
>
> **Weakness 2: It seems strange that performance doesn’t improve much after a step or two of refinement even though there’s so much headroom. Why is this? Maybe annotations should be adjusted or focused on incorrect ones? Or doing step-by-step correction is necessary?**
>
> **Response 2:** We thank Reviewer NmPr for this insightful observation. We have analyzed this phenomenon in **Section 4.4 (Table 5)** and attribute the performance plateau to the nature of the remaining errors rather than the refinement strategy itself. The VSR process is highly efficient at correcting "soft" errors, such as data omission, misalignment, or hallucination, which constitute the majority of initial mistakes. In the first round of refinement, VSR successfully identifies and corrects 92.3% of these errors, essentially maximizing the corrective potential for errors that are perceptible to the model. The errors that persist after these initial rounds stem from "hard" perception bottlenecks inherent to the base model's vision encoder. These are cases where the model fundamentally fails to resolve the fine-grained visual features due to the resolution or capacity limits of the vision backbone. in such instances, simply adjusting the annotation strategy or iterating further does not help because the model is effectively "blind" to those specific details at a feature level. Therefore, we believe the current refinement strategy is effective, and resolving these remaining stubborn errors would primarily require replacing the underlying vision backbone with a stronger model.
>
> ---
>
> If you still have any concerns or aspects you would like to discuss further, please do not hesitate to contact us at any time.
>
> We sincerely thank you for your time and thoughtful comments. If our response has addressed your concerns, we would deeply appreciate your consideration of raising your rating. We greatly value your feedback and, regardless, sincerely appreciate your engagement with our work.
>
> Best regards,
>
> The Authors

---

### Meta-Review · Area_Chair_Dj3Z · 2025-12-20

**Summary:**

The main concerns that influence my decision were missing comparisons to newer visual reasoning models like Qwen3 and InternVL.
There was an argument that the overall approach might be a narrow task specific recipe that will become obsolete with stronger base models.

**Reviewer Concerns:**

A table of results with comparisons to Qwen3 and InternVL and Gemini 2.5 address the key concern in my view.
The results remain favourable to the proposed method.

The authors counter the generality and longevity criticism by framing the proposed VSR approach as a system-2 like self verification technique as opposed to a temporary fix to a minor deficiency of a model.

**Reviewer Scores:**

The authors have asserted that there was a score change from 6 -> 8 and this appears to be substantiated by the available discussion thread from the associated reviewer. The author giving a 4 also indicated that they would increase their score.

With these changes the paper is well within the acceptance zone and the AC recommends acceptance.

---

### Decision · Program_Chairs · 2026-01-26

Accept (Poster)